# Assimilation of snow cover and snow depth into a snow model to estimate snow water equivalent and snowmelt runoff in a Himalayan catchment

Emmy E. Stigter[1], Niko Wanders[2], Tuomo M. Saloranta[3], Joseph M. Shea[4,5], Marc F.P. Bierkens[1,6], Walter W. Immerzeel[1]

[1] Department of Physical Geography, Utrecht University, The Netherlands
[2] Department of Civil and Environmental Engineering, Princeton University, United States
[3] Norwegian water resources and energy directorate (NVE), Oslo, Norway
[4] International Centre for Integrated Mountain Development, Kathmandu, Nepal
[5] Centre for Hydrology, University of Saskatchewan, Canada
[6] Deltares, Utrecht, The Netherlands

*Correspondence to*: Emmy E. Stigter (e.e.stigter@uu.nl)

**Abstract.** Snow is an important component of water storage in the Himalayas. Previous snowmelt studies in the Himalayas have predominantly relied on remotely sensed snow cover. However, snow cover data provides no direct information on the actual amount of water stored in a snowpack i.e. the snow water equivalent (SWE). Therefore, in this study remotely sensed snow cover was combined with in situ observations and a modified version of the seNorge snow model to estimate (climate sensitivity of) SWE and snowmelt runoff in the Langtang catchment in Nepal. Snow cover data from Landsat 8 and the MOD10A2 snow cover product were validated with in situ snow cover observations provided by surface temperature and snow depth measurements resulting in classification accuracies of 85.7% and 83.1% respectively. Optimal model parameter values were obtained through data assimilation of MOD10A2 snow maps and snow depth measurements using an Ensemble Kalman filter. Independent validation of simulated snow depth and snow cover with observations show improvement after data assimilation compared to simulations without data assimilation. The approach of modelling snow depth in a Kalman filter framework allows for data-constrained estimation of snow depth rather than snow cover alone and this has great potential for future studies in complex terrain, especially in the Himalayas. Climate sensitivity tests with the optimized snow model revealed that snowmelt runoff increases in winter and early melt season (December to May) and decreases during the late melt season (June to September) as a result of the earlier onset of snowmelt due to increasing temperature. At high elevation a decrease in SWE due to higher air temperature is (partly) compensated by an increase in precipitation, which emphasizes the need for accurate predictions on the changes in the spatial distribution of precipitation along with changes in temperature.

## 1 Introduction

In the Himalayas a part of the precipitation is stored as snow and ice at high elevations. This water storage is affected by climate change resulting in changes in river discharge in downstream areas (Barnett et al., 2005; Bookhagen and Burbank, 2010; Immerzeel et al., 2009, 2010). The Himalayas and adjacent Tibetan Plateau are important water towers, and water generated here supports the water demands of more than 1.4 billion people through large rivers such as the Indus, Ganges, Brahmaputra, Yangtze and Yellow (Immerzeel et al., 2010). So far, the main focus has been on the effect of climate change on the glaciers and the resulting runoff. However, snow is an important short-term water reservoir in the Himalayas, which is released seasonally contributing to river discharge (Bookhagen and Burbank, 2010; Immerzeel et al., 2009). The contribution of snowmelt to total runoff is highest in the western part of the Himalayas and lower in the eastern and central Himalayas (Bookhagen and Burbank, 2010; Lutz et al., 2014).

Although Himalayan snow storage is important for the water supply in large parts of Asia, in situ observations of snow depth are sparse throughout the region. Many studies benefit from the continuous snow cover data retrieved from satellite imagery to estimate snow cover dynamics or contribution of snowmelt to river discharge (Bookhagen and Burbank, 2010; Gurung et al., 2011; Immerzeel et al., 2009; Maskey et al., 2011; Wulf et al., 2016). Studies about snowmelt in the Himalayas have predominantly relied on remotely sensed snow cover and a modelled melt flux estimating melt runoff resulting from this snow cover (e.g. Bookhagen and Burbank, 2010; Immerzeel et al., 2009; Tahir et al., 2011; Wulf et al., 2016). However, this approach provides no or limited information on snow water equivalent (SWE), which is an important hydrologic measure as it indicates the actual amount of water stored in a snowpack. SWE can be reconstructed based on integrating a simulated melt flux over the time period of remotely sensed observed snow cover. However, this method provides only information on the peak SWE value and introduces errors when snowfall occurs during the melt season (Durand et al., 2008; Molotch, 2009; Molotch and Margulis, 2008). Currently there is only limited reliable information available on SWE for the Himalayas (Lutz et al., 2015; Putkonen, 2004). SWE can be retrieved with passive microwave remote sensing, but the results are highly uncertain, especially for mountainous terrain and wet snow (Dong et al., 2005). In addition the spatial resolution is coarse and therefore inappropriate for catchment scale studies in the Himalayas. Estimating both the spatial and temporal distribution of SWE and snowmelt is important for flood forecasting, hydropower and irrigation in downstream areas.

Selection of a suitable snow model is critical to correctly represent snow cover and SWE. Snow models of different complexity exist and can be roughly divided into physical-based and temperature-index models. Several studies have compared snow models of different complexity and their performance. Physical-based models typically outperform temperature-index models for snowpack runoff simulation on a sub-daily timescale (Avanzi et al., 2016; Magnusson et al., 2011; Warscher et al., 2013). However, physical-based and temperature-index models have similar ability to simulate daily snowpack runoff (Avanzi et al., 2016; Magnusson et al., 2015). Avanzi et al. (2016) showed that the use of a temperature-index model does not result in a significant loss of performance in simulation of SWE and snow depth with respect to a

physical-based model. Even though physical-based models outperform temperature-index models in some cases, temperature-index models are often preferred as data requirements and computational demand are lower. Especially in the Himalaya, data availability constrains the choice of a snow model.

Assimilation of remotely sensed snow cover and ground-based snow measurements have been proved to be an effective method to improve hydrological and snow model simulations (Andreadis and Lettenmaier, 2006; Clark et al., 2006; Leisenring and Moradkhani, 2011; Liu et al., 2013; Nagler et al., 2008; Saloranta, 2016). Although different data assimilation techniques exist, Kalman Filters techniques are often selected, due to their relatively low computation demand. They estimate the most likely solution using an optimal combination of observations and model simulations. Especially in catchments with strong seasonal snow cover, assimilation of remotely sensed snow cover is expected to be most useful as a result of fast changing conditions in the melting season (Clark et al., 2006).

The aim of this study is to estimate SWE and snowmelt runoff in a Himalayan catchment by assimilating remotely sensed snow cover and in situ snow depth observations into a modified version of the seNorge snow model (Saloranta, 2012, 2014, 2016). Climate sensitivity tests are subsequently performed to investigate the change of SWE and snowmelt runoff as result of changing air temperature and precipitation. The approach of modelling snow depth allows to validate the quantity of simulated snow rather than snow cover alone and is a new approach in Himalayan snow research.

## 2 Methods and data

### 2.1 Study area

The study area is the Langtang catchment, which is located in the central Himalaya approximately 100 km north of Kathmandu (Figure 1). The catchment has a surface area of approximately 580 km$^2$ from the outlet near Syabru Besi upwards. The elevation ranges from 1406 m above sea level (asl) at the catchment outlet to 7234 m asl for Langtang Lirung, which is the highest peak in the catchment. The climate is monsoon-dominated and 68% to 89% of the annual precipitation falls during the monsoon (Immerzeel et al., 2014). Spatial patterns in precipitation are seasonally contrasting, and there is a strong interaction between the orography and precipitation patterns. At the synoptic scale, monsoon precipitation decreases from south to north, but at smaller scales local orographic effects associated to the aspect of the main valley ridges (Barros et al., 2004) determine the precipitation distribution. Numerical weather models suggest that monsoon precipitation mainly accumulates at the southwestern slopes near the catchment outlet at low elevation, while winter precipitation mainly accumulates along high-elevation southern-eastern slopes (Collier and Immerzeel, 2015). Winter westerly events can also provide significant snowfall. Snow cover has strong seasonality with extensive, but sometimes erratic, winter snow cover and retreat of the snowline to higher elevations during spring and summer. For the upper part of the catchment (upstream of Kyangjin) it has been estimated that snowmelt contributes up to 40% of total runoff (Ragettli et al., 2015).

## 2.2 Calibration and validation strategy

Remotely sensed snow cover, in situ observations, and a modified version of the seNorge snow model were combined to estimate SWE and snowmelt runoff dynamics. The remotely sensed snow cover (Landsat 8 and MOD10A2 snow maps) was first validated with in situ snow cover observations provided by surface temperature and snow depth measurements. The snow model was used to simulate daily values of SWE and runoff and was forced by daily in situ meteorological observations of precipitation, temperature and incoming shortwave radiation. MOD10A2 snow cover and snow depth measurements were assimilated to obtain optimal model parameter values using an Ensemble Kalman Filter (EnKF; Evensen, 2003). The optimized parameters were used for a simulation without assimilation of the observations (open-loop). Finally, the model outcome was validated with observed snow depth and Landsat 8 snow cover.

## 2.3 Datasets

### 2.3.1 Remotely sensed snow cover

### MOD10A2

MOD10A2 is a Moderate Resolution Imaging Spectroradiometer (MODIS) snow cover product available at http://reverb.echo.nasa.gov/. The online sub-setting and reprojection utility was used to clip and project imagery for the Langtang catchment. MOD10A2 provides the 8-day maximum snow extent with a spatial resolution of ~500 m. If there is one snow observation within the 8-day period then the pixel is classified as snow. The 8-day maximum extent offered a good compromise between the temporal resolution and the interference of cloud cover. The snow mapping algorithm used is based on the Normalized Difference Snow Index (NDSI; Hall et al., 1995). The NDSI is a ratio of reflection in short-wave infrared (SWIR) and green light (GREEN) and takes advantage of the properties of snow i.e. snow strongly reflects visible light and strongly absorbs SWIR Eq. (1):

$$NDSI = \frac{GREEN - SWIR}{GREEN + SWIR} \tag{1}$$

The NDSI is calculated with MODIS spectral bands 4 (0.545-0.565 μm) and 6 (1.628-1.652 μm). Pixels are classified as snow when the NDSI ≥ 0.4. Water and dark targets typically have high NDSI values, and to prevent pixels from being incorrectly classified as snow, the reflection should exceed 10% and 11% for spectral bands 2 (0.841-0.876 μm) and 4 respectively for a pixel to be classified as snow (Hall et al., 1995). A full description of the snow mapping algorithm is given by Hall et al. (2002).

### Landsat 8

Landsat 8 imagery from 15 April 2013 to 5 November 2014 was downloaded from http://earthexplorer.usgs.gov/. Cloud free scenes (10 out of 34), based on visual inspection, were used to derive daily snow maps with high spatial resolution (30 m).

For each image digital numbers were converted to top of atmosphere reflectance. For Landsat 8 the NDSI was calculated with Eq. (1) with spectral bands 3 (0.53-0.59 μm) and 6 (1.57-1.65 μm). The chosen threshold value was equal to that used for the MOD10A2 snow cover product. The NDSI has proven to be a successful snow mapping algorithm for various sensors with a threshold value around 0.4 (Dankers and De Jong, 2004). Although the spectral bands have slightly different band

widths and spectral positions, a threshold value of 0.4 gave satisfactory results when compared with in situ snow observations. In addition, the reflection in near-infrared light should exceed 11% to prevent water from being incorrectly classified as snow (Dankers and De Jong, 2004). Therefore, a pixel is classified as snow when the NDSI value ≥ 0.4 and the reflectance in near-infrared light > 11%.

### 2.3.2 In situ observations

Different types of snow and meteorological observations were available for the study period (January 2013 – September 2014; Table 1, Figure 1). Two transects of surface temperature measurements on a north and south facing slope provided information on snow cover. The 13 temperature sensors (Hobo Tidbits) were positioned on the surface and covered by a small cairn and recorded surface temperature with 10 minute sampling intervals. Snow depths were measured with sonic ranging sensors at 4 locations at 15 minute intervals. Hourly measurements of snow depth were also made at Kyangjin and

Yala Base Camp automatic weather stations (AWS K and AWS Y; Fig. 1). Hourly means (or totals) of air temperature, liquid and solid precipitation, and incoming shortwave radiation were also recorded at AWS Kyangjin (Shea et al., 2015). Air temperature data was also acquired at several locations with 10 and 15 minute recording intervals.

### 2.4 Model forcing

The snow model was forced with daily average and maximum air temperature, cumulative precipitation and average

incoming shortwave radiation for the time period January 2013 – September 2014. Hourly measurements of air temperature, precipitation and incoming shortwave radiation at AWS Kyangjin (Shea et al., 2015) were therefore aggregated to daily values. This study period was chosen based on availability of forcing data and observations. Daily temperature lapse rates were interpolated from the air temperature measurements throughout the catchment and used to extrapolate (average and maximum) daily air temperature observed at AWS Kyangjin (Figure 1). The derived temperature lapse rates agree with the

values found by Immerzeel et al. (2014). The daily observed precipitation and temperature lapse rates were corrected in the modified seNorge snow model with the correction factors $precip$ and $T_{lapse}$ respectively to account for the uncertainty related to undercatch and the derived temperature lapse rates (Table 2). Although, temperature has a strong relation with altitude and can be accurately derived from multiple weather stations at different altitudes, small differences in the temperature lapse rate (e.g. 0.001 °C/m) can result in temperature differences up to several degrees at high altitude in

Langtang due to the extreme topography (Immerzeel et al., 2014). Hence there is a need to consider a potential correction on the temperature lapse rate. A correction is also applied to the daily observed precipitation as precipitation measurements are typically biased due to wind-induced undercatch, especially for solid precipitation (Wolff et al., 2015).

Collier and Immerzeel (2015) modelled the spatial distribution of precipitation in Langtang using an interactively coupled atmosphere and glacier mass balance model (Collier et al., 2013). Their study revealed seasonally contrasting spatial patterns of precipitation within the catchment. Monthly modelled precipitation fields from this study were therefore normalized and used to distribute the observed precipitation at AWS Kyangjin. Similarly, a radiation model (Dam, 2001; Feiken, 2014) was used to extrapolate observed incoming shortwave radiation. The radiation model takes into account the aspect, slope, elevation and shading due to surrounding topography.

The model initial conditions for January 2013 (i.e. SWE and snow depth) were set by simulating year 2013 three times.

## 2.5 Modified seNorge model

The seNorge snow model (Saloranta, 2012, 2014, 2016) is a temperature-index model which requires only data of air temperature and precipitation. In addition, the seNorge snow model includes a compaction module that can be used to assimilate and validate snow depth rather than snow cover only. The low data requirements and the compaction module make the seNorge snow model suitable for application in this study.

The seNorge snow model was rewritten from its original code into the environmental modelling software PCRaster-Python (Karssenberg et al., 2010) to allow spatio-temporal modelling of the SWE and runoff within the catchment. The snow is modelled as a single homogeneous layer with a spatial resolution of 100 m and a daily time step. The seNorge model was further improved by implementing a different melt algorithm, albedo decay and avalanching. These novel model components are described hereafter and the model parameters used are given in Table 2.

### 2.5.1 Water balance and snowmelt

Precipitation in the model is partitioned as rain or snow based on an air temperature threshold $thr_{snow}$ (°C). The snowpack consists of a solid component and possibly a liquid component. Meltwater and rain can be stored within the snowpack until its water holding capacity is exceeded and has the possibility to refreeze within the snowpack. The original melt algorithm of the seNorge snow model is substituted by the Enhanced Temperature-Index approach (Pellicciotti et al., 2005, 2008). When air temperature ($T$; °C) exceeds the temperature threshold for melt onset ($TT$; °C) the potential melt ($M_{pot}$; mm d$^{-1}$) is calculated for each pixel by Eq. (2):

$$M_{pot} = T * TF + SRF * (1 - \alpha) * R_{inc} , \tag{2}$$

where $SRF$ (m$^2$ mm W$^{-2}$ d$^{-1}$) is a radiative melt factor, $TF$ (mm °C$^{-1}$ d$^{-1}$) is a temperature melt factor, $\alpha$ (-) is the albedo of the snow cover and $R_{inc}$ (W m$^{-2}$) is the incoming shortwave radiation. In case that the threshold temperature is negative, the

potential melt can become negative when the radiation melt component is not positive enough to compensate for the negative temperature melt component. When the potential melt is negative it is set to zero to prevent negative values.

The simulated runoff in the seNorge snow model is the total runoff, i.e. the sum of snowmelt and rain. As the focus of this study is on snowmelt runoff it is necessary to split the runoff in snowmelt and rain runoff. Meltwater and rain fill up the snowpack until its water holding capacity is exceeded. The surplus is defined as snowmelt and rain runoff respectively. If both rain and snowmelt occur it is assumed that rain saturates the snowpack first. Rain falling on snow-free portions of the basin is included in the rain runoff totals.

### 2.5.2 Albedo decay

Decay of the albedo of snow is calculated with the algorithm developed by Brock et al. (2000) in which the albedo is a function of cumulative maximum daily air temperature $T_{\max}$ (°C). When $T_{\max}$ is above 0 °C the air temperature is summed as long as snow is present and no new snow has fallen. When $T_{\max}$ is below 0 °C the albedo remains constant. Albedo decay is calculated differently for deep snow (SWE ≥ 5 mm) and shallow snow (SWE < 5 mm). The albedo decay for deep snow is a logarithmic decay whereas the decay for shallow snow is exponential. This results in a gradual decrease of the albedo for several weeks, which agrees with reality (Brock et al., 2000). When new snow falls the albedo is set to its initial value. In Langtang the observed albedo of fresh snow is 0.84 and the observed minimum precipitation rate to reset the snow albedo is 1 mm d[-1] (Ragettli et al., 2015).

### 2.5.3 Avalanching

After snowfall events, avalanching occurs regularly on steep slopes in the catchment. Therefore snow transport due to avalanching is considered to be an important process for redistribution of snow in the Langtang catchment (Ragettli et al., 2015). Snow avalanching is implemented in the model using the SnowSlide algorithm (Bernhardt and Schulz, 2010). For each cell a maximum snow holding depth $SWE_{max}$ (m), depending on slope $S$ (°), is calculated using an exponential regression function following Eq. (3):

$$SWE_{max} = SS_1 + e^{-SS_2 * S},$$ (3)

where $SS_1$ and $SS_2$ are empirical coefficients. If SWE exceeds $SWE_{max}$ and the slope exceeds the minimum slope $S_{min}$ for avalanching to occur, then snow is transported to the adjacent downstream cell. Snow can be transported through multiple cells within one time step.

As the snowpack is divided into an ice and liquid component, both the ice and liquid component should be transported downwards. Avalanches in Langtang catchment mainly occur at high elevations where temperatures are low and (almost) no liquid water is present in the snowpack. It is therefore assumed that avalanches are dry avalanches and that no

liquid water is present in the avalanching snow. When there is, in rare circumstances, liquid water present in avalanching snow, the liquid water is converted to the ice component to ensure water balance closure.

### 2.5.4 Compaction and density

The compaction module is described in detail in Saloranta (2014, 2016). In this module SWE is converted into snow depth. Change in snow depth occurs due to melt, new snow and viscous compaction. The change in snow depth due to new snow is adapted such that an increase in snow depth can occur due to both snowfall and deposition of avalanching snow. The increase in snow depth due to deposition of avalanching snow is calculated using a constant snow density for dry avalanches (200 kg m$^{-3}$; Hopfinger, 1983).

### 2.6 Data assimilation

### 2.6.1 Sensitivity analysis

In order to assess which model parameters to calibrate, a local sensitivity analysis was performed by varying the value of one parameter at a time while holding the values of other parameters fixed. This gives useful first order estimates for parameter sensitivity, although it cannot account for parameter interactions. Plausible parameter values were based on literature (Table 2). The model was run in Monte-Carlo (MC) mode with 100 realizations for each parameter. The values for the parameters were randomly chosen from a uniform distribution with defined minimum and maximum values for the parameters. The snow extent and snow depth were averaged over the study period and study area for the sensitivity analysis. The sensitivity of the modelled mean snow extent and mean snow depth were compared to the changes in parameter values. A pixel is determined to be snow covered in the model when the simulated SWE exceeds 1 mm. All the parameters were varied independently per run, except for the melt factors ($TF$ and $SRF$) as these are known to be dependent on each other (Ragettli et al., 2015). Therefore, $TF$ and $SRF$ were varied simultaneously in the sensitivity analysis using a linear relation between these melt factors.

### 2.6.2 Parameter calibration

Using the Ensemble Kalman filter (EnKF; Evensen, 1994) data assimilation of snow extent and snow depth observations was used to calibrate model parameters using the framework developed by Wanders et al. (2013). Both the EnKF and particle filter (PF) have been used in several studies to assimilate snow observations into snow models (e.g. Charrois et al., 2016; Leisenring and Moradkhani, 2011; Liu et al., 2013; Magnusson et al., 2016). The EnKF and PF are similar in their approach (estimate the model uncertainty from the particle or ensemble spread). The EnKF can only be used for assimilation of continuous values and not for binary values (i.e. snow cover present or not). Therefore it is required to assimilate snow extent (continuous values) into the model, which results in a partly loss of spatial information of snow cover. However, the EnKF has a higher efficiency when it deals with Gaussian data and related errors. The computational demand required for a

PF exceeds the EnKF's computer requirements, due to the need to cover the entire (non-Gaussian) distribution. When the number of particles becomes too low, there is an additional risk of particle collapse, especially when one wants to take into account all the grid cells in the simulation with or without snow. This would require a total particle number exceeding the total number of grid cells in the domain, in combination with all the possible parameter combinations to avoid collapse of the filter. For a single site or small sites a PF would be a good alternative (e.g. Charrois et al., 2016; Magnusson et al., 2016), but limited by the current available computational power, this is only feasible with an EnKF implementation. As we deal with continuous values, it is computationally efficient and allows for dual state-parameter estimations. The lower number of ensemble members compared to a PF allowed to run multiple simulations over longer time periods, providing a better estimate of the potential of the EnKF improvements.

An advantage of the EnKF calibration framework is that it allows obtaining an uncertainty estimate for the calibrated parameters. The EnKF obtains the simulation uncertainty by using a MC framework, where the spread in the ensemble members represent the combined uncertainty of parameters and input data. Unfortunately, the EnKF does not allow to reduce and estimate the model structure uncertainty, since it relies on the assumption that the ensemble members are normally distributed. This assumption is no longer valid if multiple model schematizations are used. Therefore, it is assumed that the model is capable to accurately simulate the processes, when provided with the correct parameters. Besides the parameter and model uncertainty there is uncertainty in the observations which are assimilated. The EnKF finds the optimal solution for the model states and parameters, based on the observations and modelled predicted values and their respective uncertainties. With sufficient observations the parameters will convert to a stable solution with an uncertainty estimate that is dependent on the observations error and the ability of the model to simulate the observations. It was found that 50 ensemble members are sufficient to obtain stable parameter solutions and correctly represent the parameter uncertainty.

The EnKF was applied for each time step that observations were available. The MOD10A2 snow extent was divided into 6 elevation zones. The snow extent per elevation zone was derived from the MOD10A2 snow cover and used for assimilation to include more information on spatial distribution of snow. The elevation zone breakpoints are at 3500, 4000, 4500, 5000 and 5500 m asl. Snow maps with more than 30% cloud cover and with obvious miss-classification of snow were exempted from assimilation (3 snow maps out of 88). Only for cloud free pixels comparisons were made between modelled and observed snow extent. Two snow depth observation locations (Pluvio Langshisha and AWS Kyangjin; Figure 1) were also assimilated.

The EnKF framework allows for the inclusion of an uncertainty in the assimilated observations. Point snow depth measurements have high uncertainties that are related to limited representativeness of point snow depth observations in complex terrain due to local influence of snow drift (Grünewald and Lehning, 2015). For the snow depth measurements a variance of 25 cm was chosen to represent the uncertainty of point snow depth measurements. The MOD10A2 snow extent was assigned an uncertainty based on the classification accuracy (fraction of correctly classified pixels) determined with the in situ snow observations (Sect. 3.1.2 Remotely sensed snow cover). The uncertainty is dependent on the snow extent $SE$ ($m^2$), i.e. an increase in uncertainty for an increase in snow extent. To prevent the uncertainty to become zero when there is

no snow cover, the minimum variance for each zone was restricted to the average snow extent $\overline{SE}_{zone}$ (m$^2$) times the *accuracy* (-). Therefore the variance $\sigma^2$ per elevation zone is defined following Eq. (4):

$$\sigma^2 = \max((SE_{zone} * accuracy)^2, (\overline{SE}_{zone} * accuracy)^2) \tag{4}$$

The four most sensitive parameters ($TT$, $T_{lapse}$, $precip$ and $C_6$) resulting from the sensitivity analysis were optimized based on the assimilation of snow depth and MOD10A2 snow extent. The first three parameters ($TT$, $T_{lapse}$ and $precip$) influence both snow depth and snow extent, and were optimized by assimilating MOD10A2 snow extent. The fourth parameter ($C_6$) is an empirical coefficient relating viscosity to snow density and only influences snow depth. $C_6$ was optimized by assimilating

snow depth observations and taking into account the full uncertainty in the previously determined parameters. The two-step approach was chosen to restrict the degrees of freedom and to prevent unrealistic parameter estimates.

### 2.7 Climate sensitivity

Climate sensitivity tests were performed to investigate changes in SWE and snowmelt runoff as a result of temperature and precipitation changes. Climate sensitivity was tested by perturbing daily average air temperature, daily maximum air

temperature and daily cumulative precipitation using a delta-change method. Immerzeel et al. (2013) extracted temperature and precipitation trends from all available CMIP5 simulations for the emission scenario RCP 4.5 for Langtang catchment. They selected four models that ranged from dry to wet and from cold to warm. Four climate sensitivity tests were performed based on the projected changes in temperature and precipitation found by Immerzeel et al. (2013) (Table 3).

      Figure 2 shows the monthly cumulative precipitation and the average daily maximum temperature per month

measured at AWS Kyangjin for the study period. This data is also available for the time period 1988-2009 and is used to characterize the climatology of the catchment. Comparison of the measurements of the 1988-2009 period and the study period shows that the maximum temperature is similar for both time periods, whereas more variability exists in the cumulative precipitation. Especially in October a large difference exists in cumulative precipitation, which is caused by a large precipitation event of approximately 100mm during the study period.

**3 Results and discussion**

### 3.1 Validation of snow maps with in situ observations

### 3.1.1 In situ snow observations

Surface temperature is an indirect measure of presence of snow. Figure 3 shows observed surface temperature for two locations. Snow cover is distinguishable based on the low diurnal variability in surface temperature when snow is present

due to the isolating effect of snow (Lundquist and Lott, 2008). An optimal threshold for distinguishing between snow/no

snow was determined to be 2°C difference between daily minimum temperature and maximum temperature. The use of a larger temperature interval as threshold value was explored, however as diurnal temperature variability is small during monsoon (Immerzeel et al., 2014) setting the diurnal cycle temperature threshold above 2°C may result in incorrect monsoon snow observations.

### 3.1.2 Remotely sensed snow cover

Both observed surface temperature and snow depth measurements were converted to daily and 8-day maximum binary snow cover values to validate Landsat 8 and MOD10A2 snow cover respectively. We find that the classification accuracy of MOD10A2 and Landsat 8 snow maps based on all in situ snow observations is 83.1% and 85.7% respectively. The classification accuracy is defined as the number of correctly classified pixels divided by the total number of pixels. Table 4 shows the confusion matrices. Misclassification can be a result of variability of snow conditions within a pixel and classification of ice clouds or high cirrus clouds as snow (Parajka and Blöschl, 2006). Large viewing angles, and consequently larger observation areas may also result in misclassification (Dozier et al., 2008). MOD10A2 has a lower spatial resolution than Landsat 8 which likely causes the slightly lower accuracy for the MOD10A2 snow cover product (Hall et al., 2002). Visual inspection of MOD10A2 snow maps also revealed that some clouds are erroneously mapped as snow cover.

The accuracy of MODIS daily snow cover products are reported to be 95% for mountainous Austria (Parajka and Blöschl, 2006) and 94.2% for the Upper Rio Grande Basin (Klein and Barnett, 2003). The lower accuracy presented in this study is likely a result of the simplification of the 8-day composite product and more extreme relief and consequently larger spatial variability in snow cover. Besides classification errors, uncertainty in the in situ snow observations should be considered as well. For the in situ snow cover observations provided by surface temperature there are relatively many observations for which snow is not observed in situ, while the MOD10A2 and Landsat 8 snow maps indicate that snow should be present (Table 5). This may be caused by the fact that a thin snow layer may not result in sufficient isolation to reduce the diurnal temperature fluctuations for observation as snow (Lundquist and Lott, 2008). This observation bias in the temperature-sensed snow cover data would indicate that MOD10A2 and Landsat 8 snow maps possibly have even higher accuracies than presented here based on this validation approach.

### 3.2 Model calibration

The results of the sensitivity of mean snow extent and mean snow depth to parameter variability are shown in Table 2. The sensitivity analysis shows that the threshold temperature for melt onset ($TT$), precipitation bias ($precip$), temperature lapse rate bias ($T_{lapse}$) and the coefficient for conversion for viscosity ($C_6$) are the most sensitive parameters. For the snow compaction parameters, snow depth is most sensitive for changes in $C_6$ which is in agreement with Saloranta (2014). The melt parameters $SRF$ and $TF$ influence melt directly but show small sensitivity as these parameters are dependent on each

other. A higher value for $TF$ coincides with a lower value for $SRF$ where the value of both parameters is climate zone dependent (Ragettli et al., 2015).

Only the four most sensitive parameters were chosen to be calibrated by the EnKF to limit the degrees of freedom and to prevent the absence of convergence in the solutions for the parameters. Table 6 shows the prior and posterior
parameter distribution resulting from the assimilation of snow extent per zone and snow depth. The parameter values for $T_{lapse}$, $precip$, and $C_6$ show a narrow posterior distribution (i.e. small standard deviation) indicating that parameter uncertainty is small. $T_{lapse}$ and $precip$ represent measurement uncertainties of the model inputs. After calibration the modelled precipitation is increased and the temperature lapse rate is slightly steeper (more negative) than derived. The calibrated value of $TT$ shows a large standard deviation indicating absence of convergence in parameter solutions. This can
be either a result of insufficient data to determine the parameter value or insensitivity of the model to the parameter value. A negative value for $TT$ is plausible as melt can occur with air temperatures below 0 °C when incoming shortwave radiation is sufficient. Especially at low latitudes and high elevation, solar radiation is an important cause of snowmelt (Bookhagen and Burbank, 2010). $TT$ is reported to be as negative as -6 °C for Pyramid, Nepalese Himalayas (Pellicciotti et al., 2012). Here $TT$ lies in a range which is even more negative than -6 °C. This is likely to be partly a result of the model structure. When
$TT$ is negative the melt algorithm (Eq. (2)) can give negative values. The temperature term in Eq. 2 becomes negative in case the air temperature is below zero degrees but higher than TT. The reason for negative melt to occur in a few rare cases is a limitation of the EnKF calibration in combination with the Enhanced Temperature-index method. The EnKF does not allow to constrain parameter ranges and this results in a relative low TT, which may occasionally lead to negative melt when incoming shortwave radiation is low and the air temperature is above TT. In those cases when negative melt occurs it is
capped to zero and as a results the model is relatively insensitive for low temperatures close to the TT and the EnKF does not converge into a parameter solution.

### 3.3 Model validation

#### 3.3.1 Snow cover

Both the modelled and MOD10A2 snow extent show strong seasonality of snow cover in the catchment (Figure 4). After
calibration modelled snow extent shows notable improvement in elevation zone 3500-4000 m asl during the melt season in 2014. After calibration the threshold temperature for melt onset is lower, resulting in more and earlier onset of snowmelt. Consequently there is a decreased snow extent. The zones in the lower areas are expected to show most improvement as this is the area where snow cover is ephemeral and considerable improvements of the modelled snow extent in elevation zone 3500-4000 m asl are indeed observed (Figure 4). The root mean square error (RMSE) decreased from 14.2 to 11.2 km$^2$ after
calibration. The simulated snow extent agrees well with MOD10A2 observed snow cover for the higher elevation zones (>4500 m asl). An exception is the snow extent in summer 2013 in the elevation zone 5000-5500 m asl. The snow model underestimates the snow extent compared to the MOD10A2 snow extent. This discrepancy is possibly the result of i)

overestimation of simulated melt, ii) an actual snow event that is simulated as rain by the model due to too high air temperature, or iii) erroneous mapping of clouds as snow in the MOD10A2 snow cover.

The model classification accuracy of snow cover after calibration is 85.9% based on pixel comparison between modelled 8-day maximum snow extent and MOD10A2 snow extent. The classification accuracy is the average classification accuracy over all members. There is only a slight increase of 0.2% in accuracy after calibration, however the performance was already high (85.7%) before calibration. The classification accuracy is lower on steep slopes where avalanching is common, and as the snow extent in avalanching zones is highly dynamic this is not well captured in the model. Calibration of parameters that influence avalanching might overcome this discrepancy to some degree, however a more advanced approach to avalanche modelling may be required. In addition the spatial resolution of the remotely sensed snow cover is likely to be insufficient to detect the avalanche dynamics. Other potential explanations for lower classification accuracies are uncertainties related to the simulated precipitation phase (rain/snow) and the simulated spatial distribution of precipitation based on Collier and Immerzeel (2015).

Landsat 8 derived snow extent is lower in winter than the modelled snow extent and the MOD10A2 snow extent (Figure 4). Distinct differences between the Landsat 8 instantaneous snow cover observations and the MOD10A2 8-day maximum snow cover extents (Figure 4) can be attributed to (i) the sensitivity of the Landsat 8 snow cover maps to misclassified snow pixels in shaded area, (ii) the much higher spatial resolution of Landsat 8 (Hall et al., 2002) and (iii) the difference between an instantaneous image and an 8-day composite.

The model classification accuracy, based on pixel comparison with Landsat 8 snow maps, increased from 74.7% to 78.2% after calibration. In Table 7 individual model classification accuracy is given based on comparison with each Landsat 8 snow map. Relative low accuracies occur in winter (especially at 20-12-2013 and 05-01-2014) and the model overestimates snow cover compared to the Landsat 8 snow maps (Figure 4). The overestimation of snow cover by the model on 20-12-2013 is particularly large and it can be explained by a small snow event (2.3 mm measured at Kyangjin) a few days before the acquisition. With below zero temperatures the model simulates a large snow cover extent, but based on a very small amount. Snow redistribution by wind, a patchy snow cover and/or sublimation may also explain the mismatch with the Landsat 8 snow cover in this particular case.

### 3.3.2 Snow depth

The observed and modelled snow depths at four locations are shown in Figure 5. The simulated snow depth is given for the model simulations i) without calibration, ii) after calibration of snow extent, and iii) after calibration of both snow extent and snow depth. After calibration with snow extent there is an increase in snow depth for Yala Pluvio and Yala BC for the entire snow season as result of increased simulated precipitation. For Langshisha and Kyangjin the snow depth mainly decreased after calibration with snow extent. These stations are at a lower elevation, and since the threshold temperature for melt onset is lowered after calibration, this leads to reduced snow depth. At all locations the modelled snow depth decreased after calibration with both snow extent and snow depth due to lowering of the parameter relating snow density to snow depth.

After calibration with both snow extent and snow depth, comparison of modelled and observed snow depth at Langshisha shows good agreement. Especially after calibration the timing of the melt onset during spring is improved. For Yala Pluvio and Yala BC the agreement between modelled and observed snow depth is also good, though improvement of the timing of melt onset is limited. For Kyangjin the modelled snow depth agrees less well with observed snow depth in spring 2013, but it

improves in 2014. In spring the snow cover duration of snow events decreases after calibration and improves the fit with the observed snow depth.

Yala Pluvio and Yala BC are the only locations that serve as an independent validation of snow depth as these stations are not used for the assimilation. The simulated melt onset in spring is later compared to what is observed. The diurnal variability of air temperature is high during the pre-monsoon season (March to mid-June; Immerzeel et al., 2014).

Though daily average air temperatures are  below zero, positive temperatures and snowmelt can occur in the afternoon above 5000 m asl (Shea et al., 2015; Ragettli et al., 2015). This can explain the difference between simulated and observed melt onset. Using an hourly time step might therefore improve the simulation of snowmelt in spring (Ragettli et al., 2015). While the timing of snowpack depletion at Yala Pluvio and Yala BC are offset from the observations, the modelled quantity of snow is in the same order of magnitude for both modelled and observed time series. Hence there is no substantial

overestimation or underestimation of snow depth. The RMSE between simulated and observed snow depth decreases after calibration with both snow extent and snow depth compared to the uncalibrated simulation of snow depth and after calibration of only snow extent. This shows the benefit of assimilating both snow extent and snow depth into the snow model to obtain optimal parameter values.

While this study shows an approach in using snow depth observations for assimilation and validation, only four

locations with snow depth observations were available. The number of available snow depth observations and the choice of different stations for assimilation might influence the results. Four snow depth observations are insufficient for systematic assimilation and independent validation. However, our approach is useful and is recommended for future studies in the Himalayas in particular when more point observations of snow depth are available.

**3.4 Climate sensitivity of SWE and snowmelt runoff**

The cumulative basin-wide mean snowfall is 1222 mm for the simulation period. Nearly one-third (31.4%) of the snowfall is transported to lower elevations due to avalanching, and 16.2% of the snowfall is transported to elevations lower than 5000 m asl. Transport of snow to lower elevations contributes to snowmelt runoff and has been estimated to be 4.5% of the total water input for the upper part of the Langtang catchment (Ragettli et al., 2015).

The simulation of the SWE for the study period shows a pattern of increasing SWE with increasing elevation (Figure 6 and Figure 7). At higher elevation air temperature is lower with more snow accumulation than melt, resulting in a higher gain in SWE over time. The glaciers Langtang and Langshisha are positioned at approximately the same elevation (Ragettli et al., 2015), though the SWE is considerably higher at Langshisha glacier (Figure 6) due to the precipitation

distribution approach we use. Also, some areas at higher elevation show less SWE than surrounding areas at the same elevation. These areas represent the steep slopes in the catchment where avalanching occurs regularly. The transported snow accumulates below these steep slopes. The simulated avalanches are based on a simple model parameterization in which the snow is transported via single stream paths, resulting in a few pixels with extreme accumulation of SWE. This is mainly visible in the northeastern part of the catchment. Modelling the divergence of transported snow might improve the extreme accumulation simulated for some pixels.

For the climate sensitivity tests a delta-change method is used. This method has limitations as climate variability of future climate is not constant compared to the study period (Kobierska et al., 2013). In addition Kobierska et al. (2013) showed that changes in runoff due to climate change is predicted differently by a physical-based snow model and a parameterized snow model for a glacierized catchment. Parameterized snow models (such as the modified seNorge snow model that is used in this study) are calibrated to fit the current climate and not future climate and might therefore be incapable of predicting future states of the snowpack. However, the scope of this study is to show the sensitivity of the SWE and snowmelt runoff to changes in air temperature and precipitation, and not a full-fledged climate impact study. Therefore the use of a parameterized snow model and the delta-change method is suitable in this case.

Figure 7 and Figure 8 show the results of the absolute and relative change in SWE for different climate sensitivity tests. All climate sensitivity tests show a decrease in SWE, but the relative change is greatest at low elevations in the valley. We also observe a strong gradient of decreased relative change in SWE with increased elevation. An increase in temperature leads to an increase in melt and more precipitation in form of rain instead of snow. Both processes result in decreased relative change of SWE with elevation. Near the catchment outlet there is an area with 100% decrease in SWE as precipitation will only fall as rain instead of snow.

A slight deviation from the elevational trend in SWE change occurs between 3000 and 4000 m asl, a zone that could be sensitive to changes in the elevation at which snowfall occurs. The combination of snowfall at higher elevations due to higher temperature and the monthly differing spatial patterns in precipitation are likely to explain the banded patterns.

Changes in SWE and the spatial distribution of SWE will also be affected by changes in total precipitation. The influence of precipitation can be determined based on comparison of the two wet and dry climate sensitivity tests (Figure 7 and Figure 8). A decrease in precipitation results in decreased SWE as there is less snowfall. However, the increased precipitation for the wet/cold and wet/warm climate sensitivity tests (+12.1 and +12.4%, respectively) does not compensate for the temperature-related increase in melt and decrease in snowfall in the valley.

Reduced warming under the wet/cold climate sensitivity test results in a smaller decrease of SWE compared to the wet/warm climate sensitivity test, even in the valley. At higher elevations changes in SWE are weakly negative and in some areas positive. Snowpack sensitivity to temperature change decreases with elevation (Brown and Mote, 2009). The increased SWE under both wet climate sensitivity tests occurs in the southeastern part of the catchment where relatively large amounts of precipitation occur in winter (Collier and Immerzeel, 2015). Schmucki et al. (2015) showed similar results for the Alps. They showed that low- and mid-elevation stations are sensitive to temperature change but not to precipitation change. In

contrast, at high-elevation stations an increase in precipitation partly compensates for an increase in temperature. The compensating effect of increased precipitation at high elevations is important for glacier systems, and emphasizes the importance of accurate estimations of both change in precipitation and its spatial distribution.

The modelled snowmelt and rain runoff at the catchment outlet is greatest during the monsoon and low during winter (Figure 9). Peak snowmelt and rain runoff occur in June and July respectively. The snowmelt season starts in March when temperatures and insolation are rising, and continues until October. Snowmelt runoff contributes most to total runoff during pre-monsoon and early-monsoon (March-June), which is in agreement with Bookhagen and Burbank (2010). Validation of the simulated runoff with observed runoff was impossible, because (i) there was no reliable runoff data available for the study period as there was no reliable rating curve and (ii) the model focusses on rain and snowmelt runoff, however glacier runoff and delay of runoff due to groundwater and glacier storage is not incorporated in the model structure.

The climate sensitivity of snowmelt and rain runoff is shown in Figure 9. All climate sensitivity tests show an increase in snowmelt runoff from October to May. In contrast, snowmelt runoff decreases from June to September. Higher temperatures result in more snowmelt and less snowfall during winter and an early melt season which leads to a shift in the peak of snowmelt runoff. In other mountain regions similar changes in runoff patterns appear. Several studies in the Alps show that the peak in snowmelt runoff will be shifted from summer to late spring (Bavay et al., 2009, 2013; Kobierska et al., 2013). Immerzeel et al. (2009) showed that in the upper Indus Basin the peak in snowmelt runoff appears one month earlier by 2071-2100 as result of an increase in temperature and precipitation. However, Immerzeel et al. (2012) showed that total snowmelt runoff remains more or less constant under positive temperature and precipitation trends in the upper part of the Langtang catchment. In their study snowmelt on glaciers is not defined as snowmelt runoff and is therefore a minor component of total runoff, leading to different results.

For the wet climate sensitivity tests total runoff (i.e. the sum of snowmelt and rain runoff) increases throughout the year. The decrease in melt runoff during late melt season is compensated by the increase in rain runoff as there is more precipitation. The future hydrology of the central Himalayas largely depends on precipitation changes as it is dominated by rainfall runoff during the monsoon (Lutz et al., 2014). As we perturb the model with a percentage change in precipitation that is constant through the year, the absolute change in precipitation is greater in the monsoon than in winter. For climate sensitivity tests with decreased precipitation, total runoff from June to September decreases, but from October to May it increases as a result of increased snowmelt. Estimates of seasonal changes in precipitation are thus critical to determine whether rain and snowmelt runoff increases or decreases during monsoon.

## 4 Conclusions

Remotely sensed snow cover, in situ observations and a modified seNorge snow model were combined to estimate (climate sensitivity of) SWE and snowmelt runoff in the Langtang catchment. Validation of remotely sensed snow cover (Landsat 8 and MOD10A2 snow maps) show high accuracies (85.7% and 83.1% respectively) against in situ snow observations

provided by surface temperature and snow depth measurements. Data assimilation of MOD10A2 snow cover and snow depth measurements using an EnKF proves to be successful for obtaining optimal model parameter values. Independent validation of simulated snow depth and snow cover against snow depth measurements and Landsat 8 snow cover show improvement after assimilation of snow depth and snow cover compared to results before data assimilation. The applied methodology of simultaneous assimilation of snow cover and snow depth allows for the calibration of important snow parameters and validation of the snow depth rather than snow cover alone. This opens up new possibilities for future snow assessments and sensitivity studies in the Himalayas.

The spatial distribution of SWE averaged over the simulation period (January 2013-September 2014) shows a strong gradient of increasing SWE with increasing elevation. In addition the SWE is considerably higher in the southeastern part of the catchment than the northeastern part of the catchment as a result of the spatial and temporal distribution of precipitation.

Finally the climate sensitivity study revealed that snowmelt runoff increases in winter and early melt season (December to May) and decreases during the late melt season (June to September) as a result of the earlier onset of snowmelt due to increasing temperature. There is a strong relative decrease in SWE in the valley with increasing temperature due to more snowmelt and less precipitation as snow. At higher elevations an increase in precipitation partly compensates for increased melt due to higher temperatures. The compensating effect of precipitation emphasizes the importance and need for accurate prediction of change in spatial and temporal distribution of precipitation.

*Data availability.* Data is available upon request to the corresponding author.

*Competing interests.* The authors declare that they have no conflict of interest.

*Acknowledgements.* This project was supported by funding from the European Research Council (ERC) under the European Union's Horizon 2020 research and innovation program (grant agreement no. 676819) and by the research programme VIDI with project number 016.161.308 financed by the Netherlands Organisation for Scientific Research (NWO). The authors thank Hendrik Wulf and two anonymous reviewers for their constructive comments that helped improving the manuscript.

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

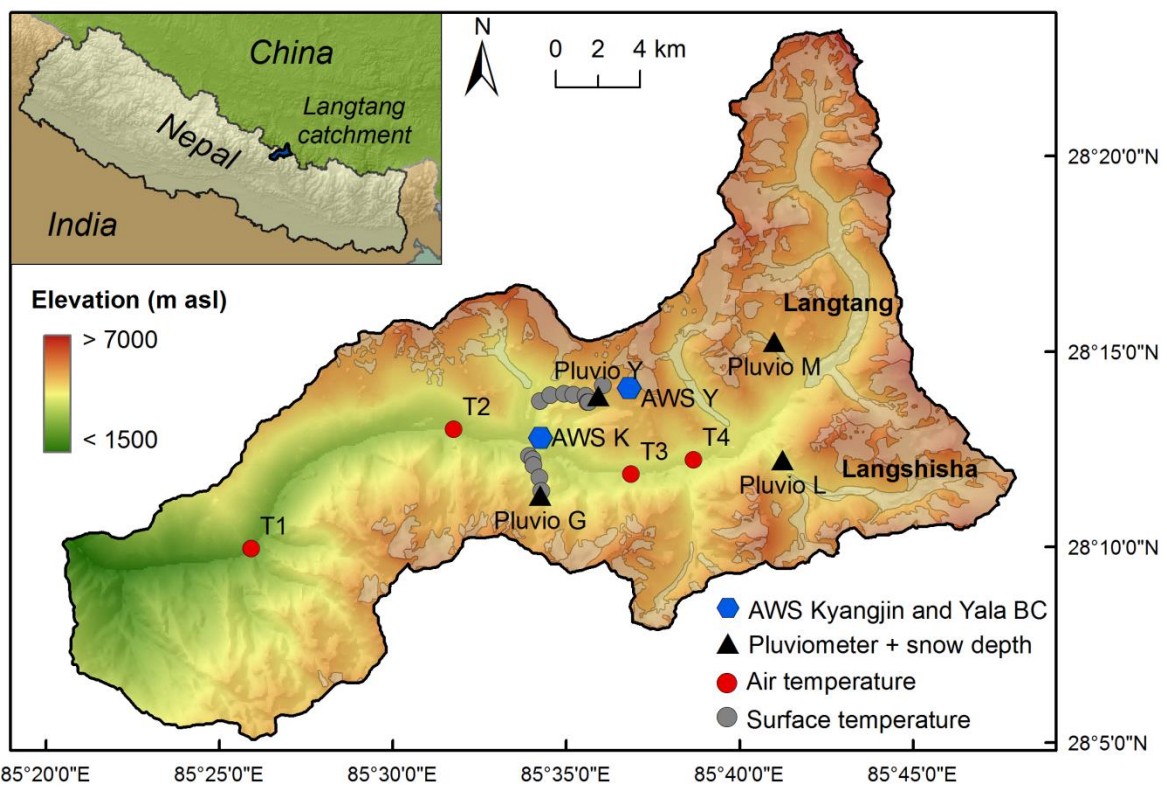

**Figure 1: Study area with the locations of the in situ observations. Langtang and Langshisha refer to the two main glaciers in the upper Langtang valley.**

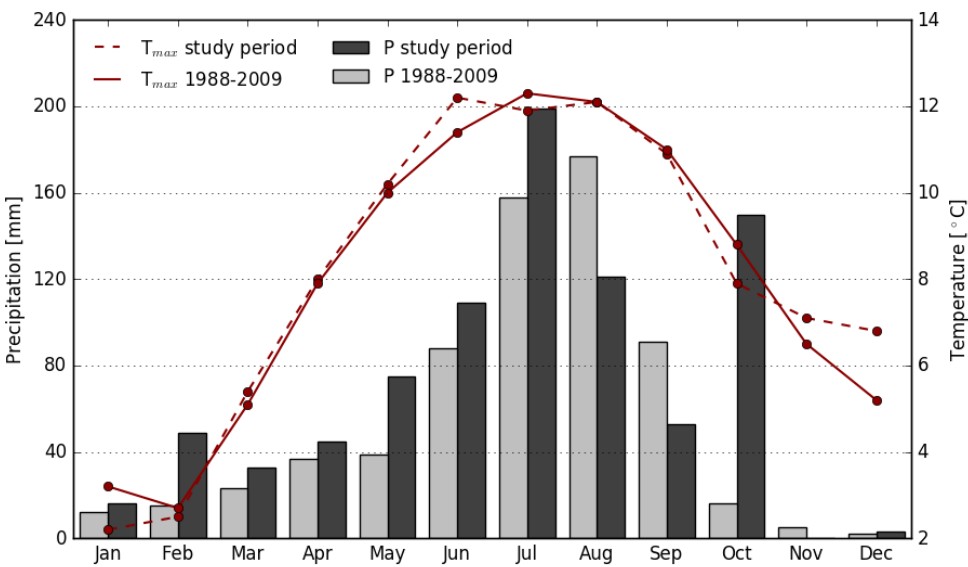

**Figure 2 Comparison of maximum temperature (T$_{max}$) and cumulative monthly precipitation (P) for the study period (January 2013-September 2014) and the 1988-2009 time series (based on measurements in Kyangjin). The average yearly cumulative precipitation is 853mm and 663mm for the study period and the 1988-2009 time series respectively.**

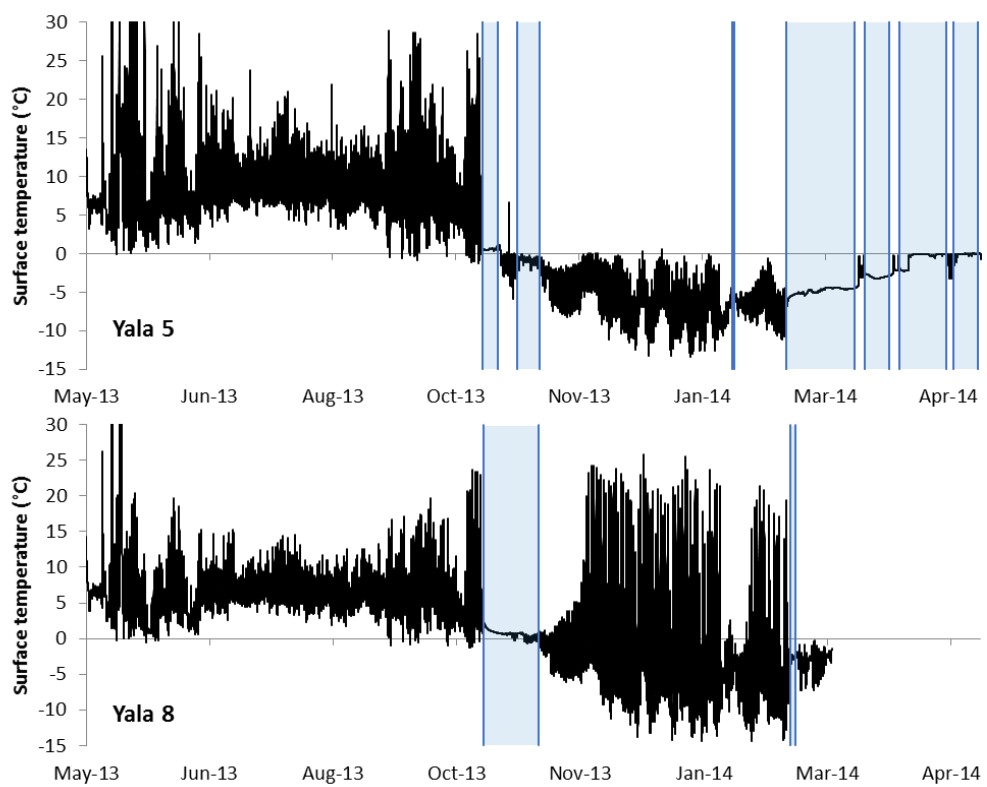

**Figure 3: Observed surface temperature with 10 minute interval at two locations (Table 1). The blue vertical lines indicate the start and end of the snow cover.**

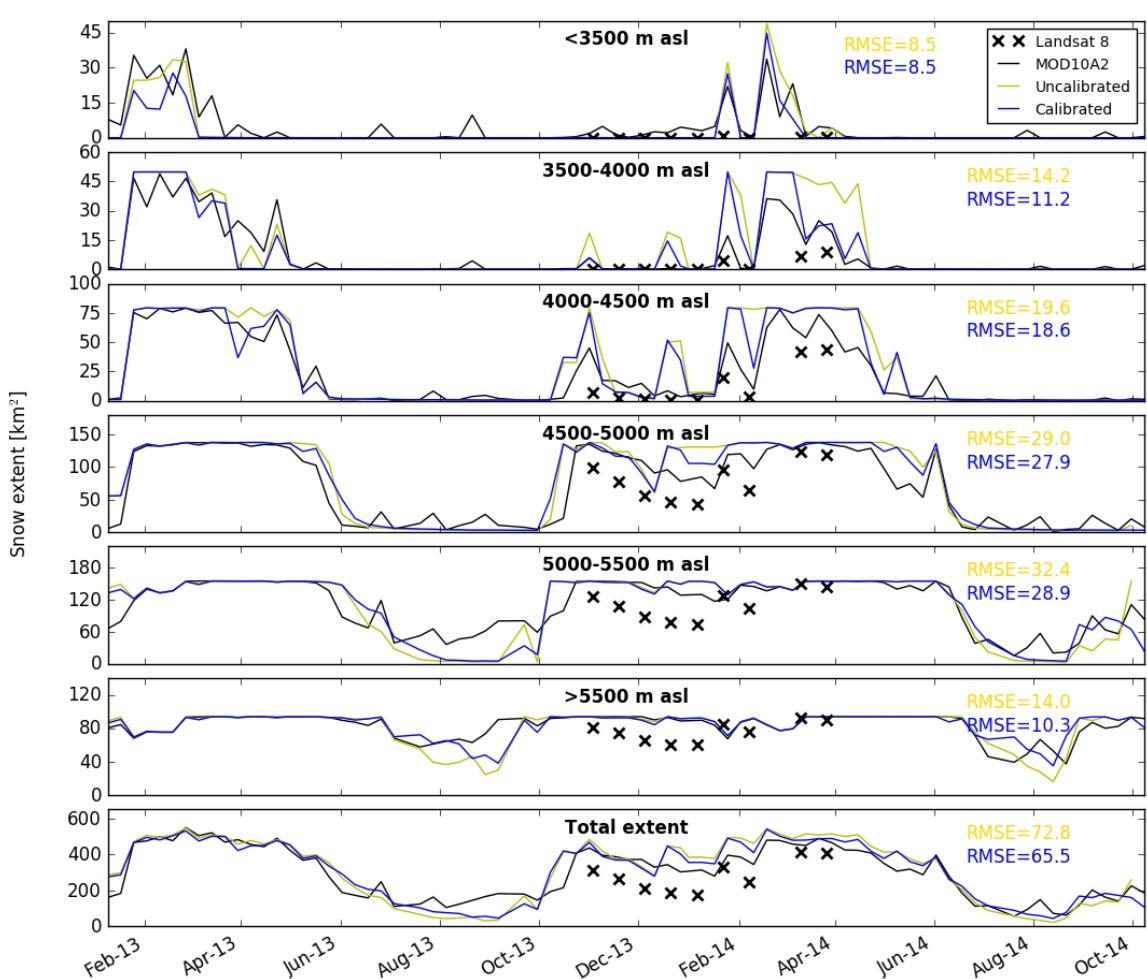

**Figure 4: Modelled 8-day maximum snow extent before and after calibration (ensemble mean), Landsat 8 snow extent and MOD10A2 snow extent per elevation zone. The RMSE (km$^2$) is given per zone for the fit between modelled (before and after calibration) and MOD10A2 snow extent.**

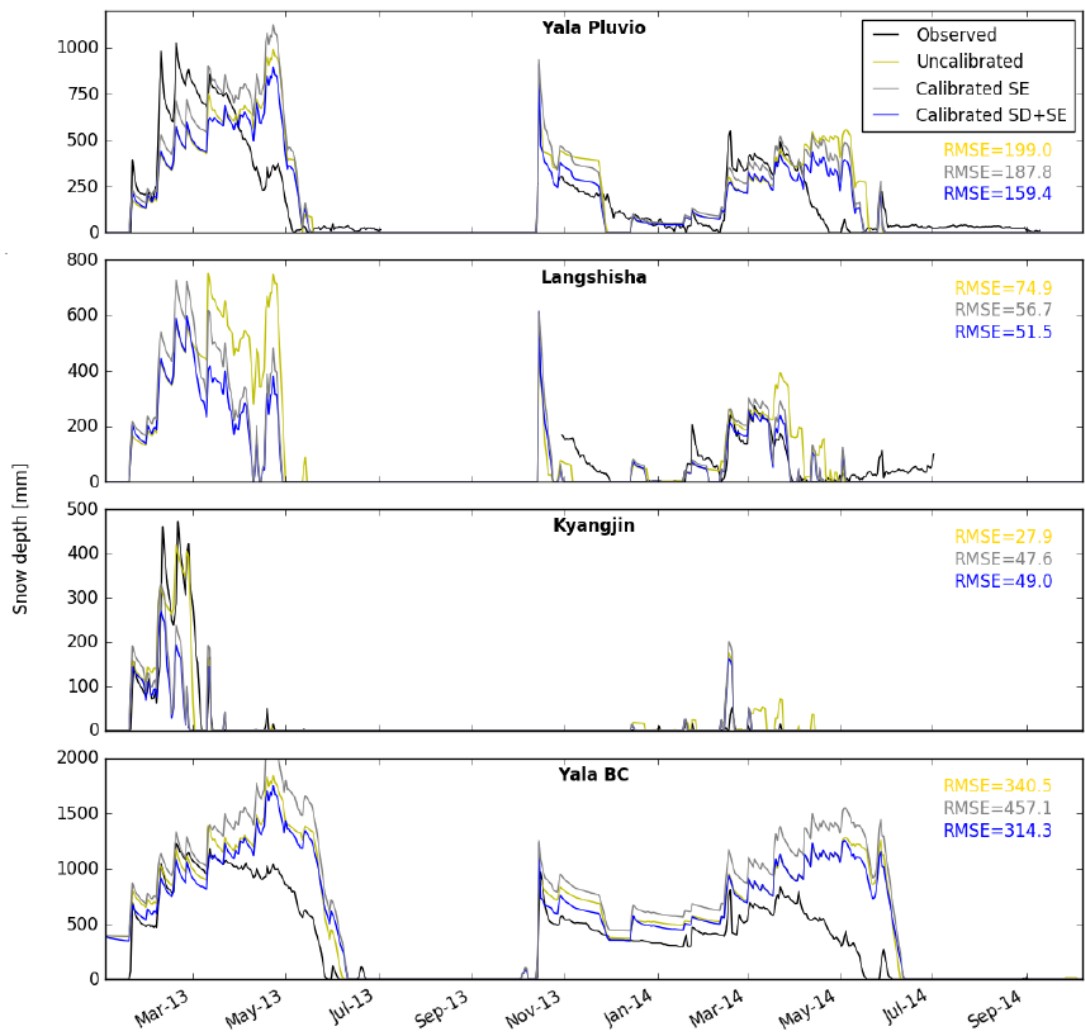

**Figure 5: Observed snow depth and modelled snow depth i) before calibration, ii) after calibration of snow extent (SE), and iii) after calibration of both snow extent and snow depth (SD+SE) (ensemble mean) at three locations. The RMSE (mm) is given for the fit between modelled (before and after calibration) and observed snow depth.**

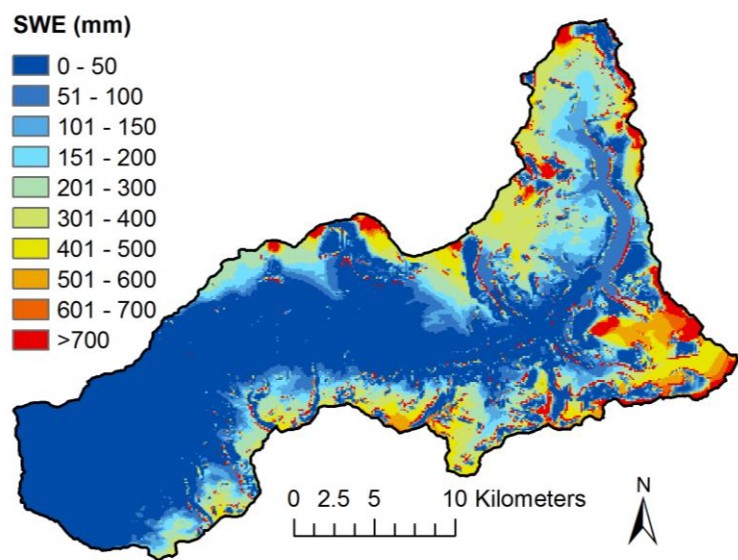

**Figure 6: Spatial distribution of ensemble mean annual average snow water equivalent (SWE).**

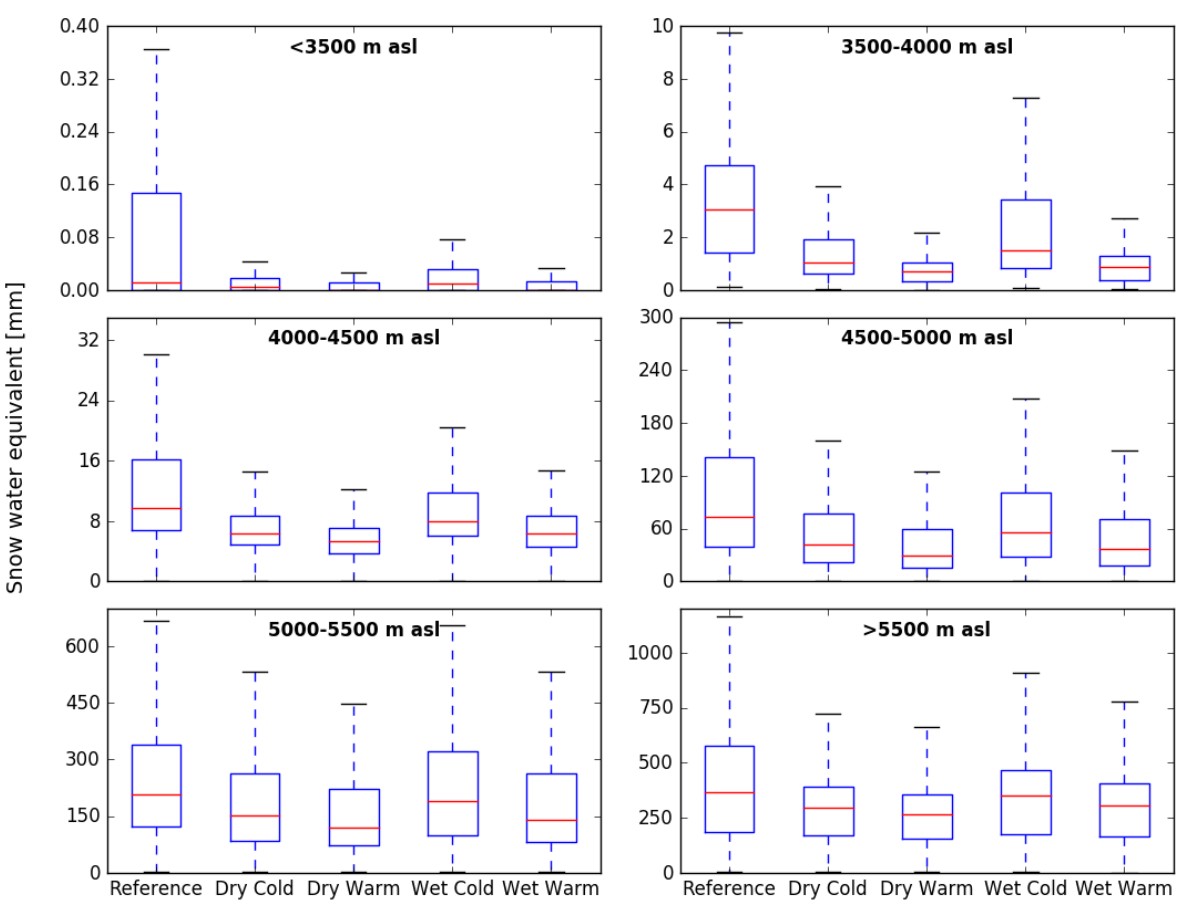

**Figure 7 Boxplots of SWE per elevation zone averaged over the simulation period and all ensemble members for the study period (reference) and the four climate sensitivity tests (Table 3).**

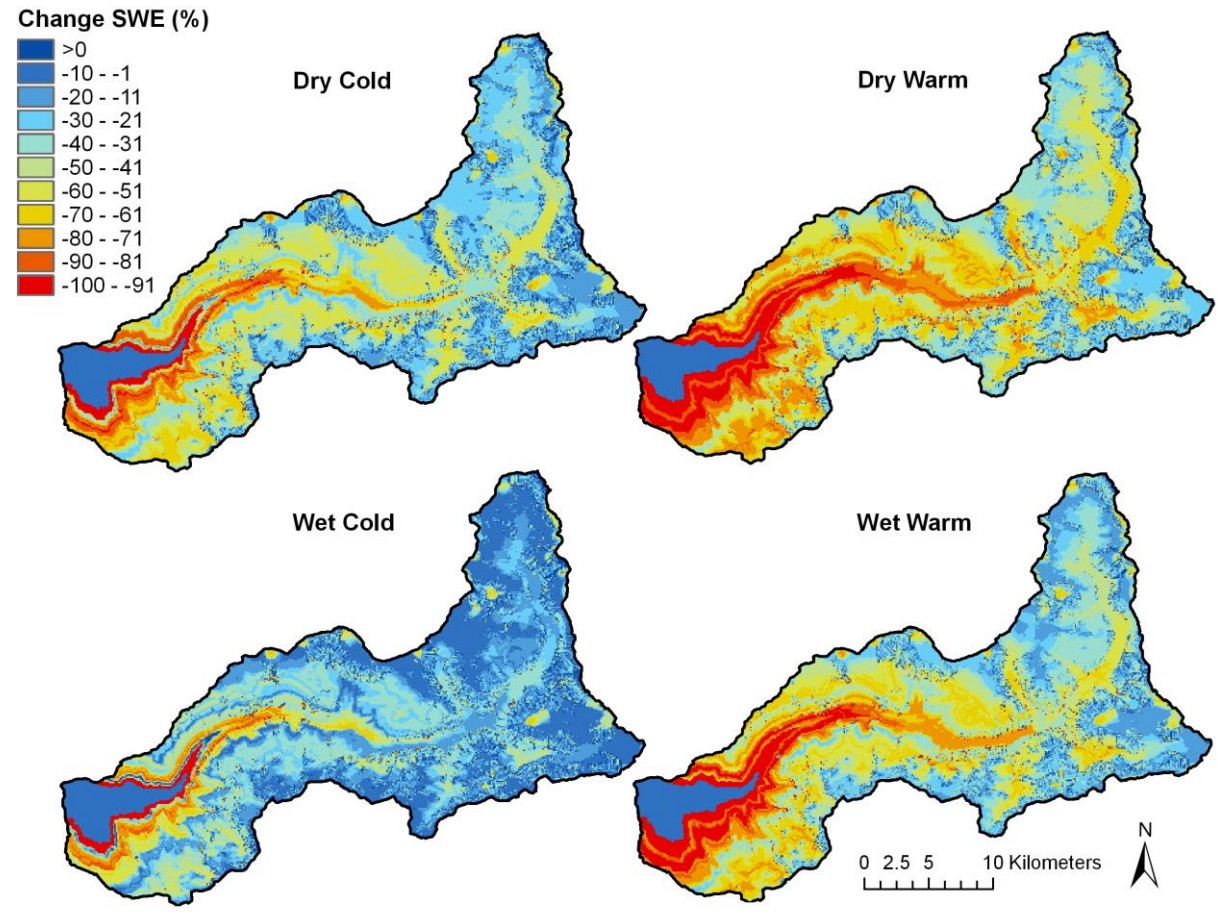

**Figure 8: Change in SWE averaged over the simulation period and all members for each climate sensitivity test (Table 3).**

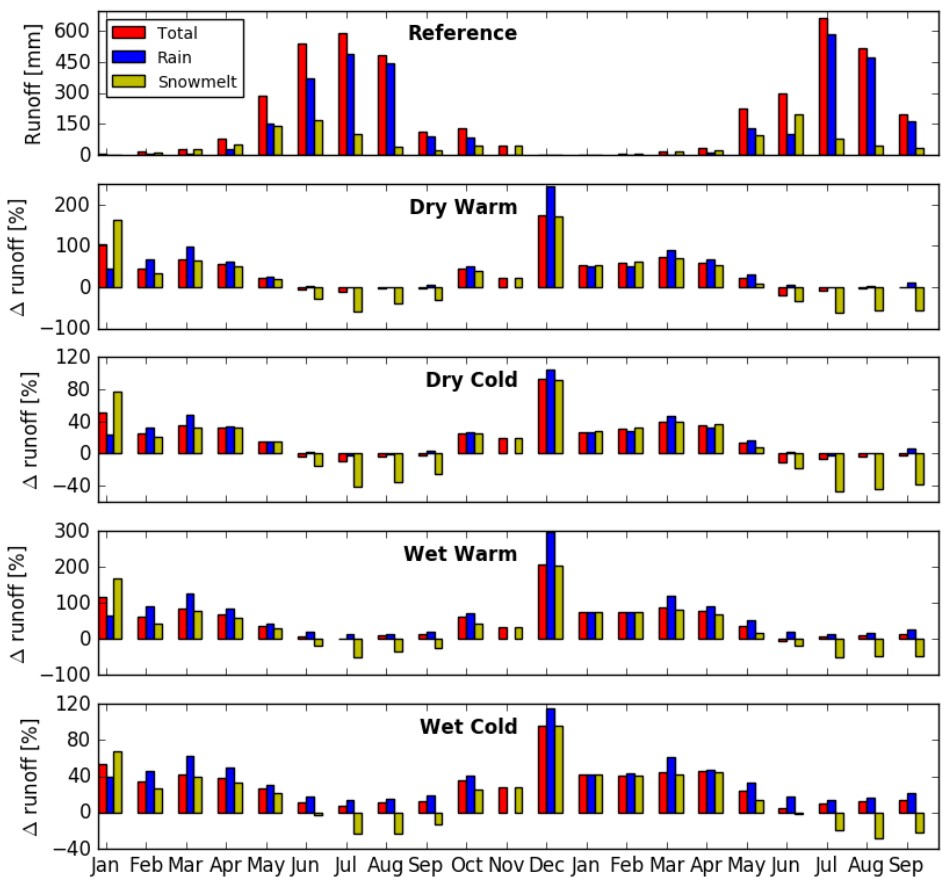

**Figure 9: Modelled runoff at catchment outlet for the study period (January 2013 – September 2014) and change in runoff compared to the study period for the climate sensitivity tests.**

**Table 1: Overview of the in situ observations and their specifications. Locations are shown in Fig. 1.**

| Description | Code | Data availability | Latitude | Longitude | Elevation (m asl) | Observations[1] |
|---|---|---|---|---|---|---|
| Yala 1 | Y1 | 06/05/13 – 03/05/14 | 28.22645 | 85.56878 | 4117 | ST |
| Yala 2 | Y2 | 06/05/13 – 03/05/14 | 28.22897 | 85.57391 | 4214 | ST |
| Yala 3 | Y3 | 06/05/13 – 03/05/14 | 28.2298 | 85.58051 | 4328 | ST |
| Yala 4 | Y4 | 06/05/13 – 02/03/14 | 28.22932 | 85.58492 | 4441 | ST |
| Yala 5 | Y5 | 06/05/13 – 03/05/14 | 28.22894 | 85.5908 | 4541 | ST |
| Yala 6 | Y6 | 06/05/13 – 03/05/14 | 28.22635 | 85.5918 | 4656 | ST |
| Yala 7 | Y7 | 06/05/13 – 02/03/14 | 28.22635 | 85.59246 | 4759 | ST |
| Yala 8 | Y8 | 06/05/13 – 02/03/14 | 28.23342 | 85.59921 | 4960 | ST |
| Ganjala 1 | G1 | 03/11/13 – 11/10/14 | 28.20305 | 85.56405 | 3908 | ST |
| Ganjala 2 | G2 | 03/11/13 – 06/09/14 | 28.20155 | 85.56577 | 3998 | ST |
| Ganjala 3 | G3 | 03/11/13 – 11/10/14 | 28.19899 | 85.56617 | 4094 | ST |
| Ganjala 4 | G4 | 03/11/13 – 30/04/14 | 28.1938 | 85.56916 | 4201 | ST |
| Ganjala 5 | G5 | 03/11/13 – 11/10/14 | 28.18831 | 85.57001 | 4300 | ST |
| Pluvio Yala | Pluvio Y | 01/01/13 – 30/06/13 26/10/13 – 16/10/14 | 28.22900 | 85.59700 | 4831 | T, SD |
| Pluvio Ganjala | Pluvio G | 20/01/14 – 03/05/14 | 28.18625 | 85.56961 | 4361 | SD |
| Pluvio Langshisha | Pluvio L | 29/10/13 – 01/07/14 | 28.20265 | 85.68619 | 4452 | SD |
| Pluvio Morimoto | Pluvio M | 17/05/13 – 09/10/14 | 28.25296 | 85.68152 | 4919 | T, SD |
| Lama Hotel | T1 | 01/01/13 – 07/10/14 | 28.16212 | 85.43073 | 2492 | T |
| Langtang | T2 | 01/01/13 – 07/10/14 | 28.21398 | 85.52745 | 3557 | T |
| Jathang | T3 | 01/01/13 – 07/10/14 | 28.1958 | 85.6132 | 3947 | T |
| Numthang | T4 | 01/01/13 – 07/10/14 | 28.20213 | 85.64313 | 3983 | T |
| AWS Kyangjin | AWS K | 01/01/13 – 07/10/14 | 28.2108 | 85.5695 | 3862 | T, SD, P, IR |
| AWS Yala Base Camp | AWS Y | 01/01/13 – 07/10/14 | 28.23252 | 85.61208 | 5090 | SD |

[1] ST, surface temperature; SD, snow depth; T, air temperature; P, precipitation; IR, incoming shortwave radiation

**Table 2: Parameters in the snow model. Value indicates the uncalibrated parameter value and the value range indicates the range which is used for the sensitivity analysis. Sensitivity of snow depth (SD) and snow extent (SE) represent the difference between the 90th and 10th percentile of mean snow depth and snow extent resulting from the sensitivity analysis.**

| Parameter | Unit | Description | Initial Value | Value range | Sensitivity SD [mm] | Sensitivity SE [km$^2$] |
|---|---|---|---|---|---|---|
| TT | [°C] | Threshold temperature for onset of melt or refreezing | 0 [6] | -6 – 2 [4, 5, 6] | 157.3 | 57.25 |
| SRF | [m$^2$ mm W$^{-2}$ d$^{-1}$] | Melt factor dependent on incoming shortwave radiation | 0.15 [6] | 0.13 – 0.19 [4, 6] | 9.486 | 2.721 |
| TF | [mm °C$^{-1}$ d$^{-1}$] | Melt factor dependent on temperature | 4.32 [6] | 2.54 – 5.19 [4, 6] | 9.486 | 2.721 |
| thr$_{snow}$ | [°C] | Threshold for partitioning in rain or snow | 0 [6] | -1 – 1 [6,7] | 35.82 | 11.99 |
| C$_{rf}$ | [mm °C$^{-1}$ d$^{-1}$] | Degree-day refreezing factor | 0.16 [7] | 0.08 – 0.40 [7] | 8.188 | 0.3248 |
| a$_{ini}$ | [-] | Decay of albedo deep snow (initial) | 0.713 [2] | - | - | - |
| α$_u$ | [-] | Albedo of surface underlying snow (ground, ice) | 0.15, 0.25 [6] | - | - | - |
| a$_1$ | [-] | Decay of albedo deep snow | 0.112 [2] | 0.112 – 0.34 [2,6] | 56.39 | 7.279 |
| a$_2$ | [-] | Decay of albedo shallow snow | 0.442 [2] | 0.3 – 0.5 | 0.2410 | 0.2818 |
| a$_3$ | [-] | Decay of albedo shallow snow (exponent) | 0.058 [2] | 0.03 – 0.1 | 0.2001 | 0.2132 |
| r$_{max}$ | [-] | Maximum allowed fraction of liquid water in snowpack | 0.1 [7] | 0.05 – 0.20 [7] | 31.66 | 0.3278 |
| d$^*$ | [cm] | Scaling length for smooth transition albedo from deep snow to shallow snow | 2.4 [2] | 1 – 25 | 0.0012 | 0.0007 |
| SS$_1$ | [m] | Regression function parameter snow holding depth dependence on slope angle | 250 [6] | 200 – 300 | 10.86 | 2.033 |
| SS$_2$ | [-] | Regression function parameter snow holding depth dependence on slope angle | 0.172 [6] | 0.16 – 0.19 | 26.45 | 7.170 |
| S$_{min}$ | [°] | Minimum slope for avalanching to occur | 25 [1] | 15 – 35 | 34.00 | 1.640 |
| ρ$_{av}$ | [kg L$^{-1}$] | Density of avalanching snow | 0.200 [3] | - | - | - |
| ρ$_{min}$ | [kg L$^{-1}$] | Minimum density of new snow due to snowfall | 0.050 [7] | 0.050 – 0.150 [7] | - | - |
| a$_{ns}$ | | Coefficient for density of new snow | 100 [7] | - | - | - |
| η$_0$ | [MN s m$^{-2}$] | Coefficient related to viscosity of snow (at zero temperature and density) | 7.6 [7] | 1 – 10 [7] | 75.75 | - |
| C$_5$ | [°C$^{-1}$] | Coefficient for temperature effect on viscosity | 0.1 [7] | 0.04 – 0.12 [7] | 10.44 | - |
| C$_6$ | [L kg$^{-1}$] | Coefficient for density effect on viscosity | 21 [7] | 15 – 35 [7] | 268.8 | - |
| k$_{comp}$ | [-] | Compaction factor | 0.5 [7] | - | - | - |
| precip | [-] | Precipitation correction factor | 1 | 0.6 – 1.4 | 320.1 | 14.17 |
| T$_{lapse}$ | [-] | Temperature lapse rate correction factor | 1 | 0.9 – 1.1 | 116.0 | 24.63 |

5    [1] Bernhardt and Schulz, 2010     [3] Hopfinger, 1983     [5] Ragettli et al., 2013     [7] Saloranta, 2014
     [2] Brock et al., 2000     [4] Pellicciotti et al., 2012     [6] Ragettli et al., 2015

**Table 3: Changes in temperature (ΔT) and precipitation (ΔP) for the climate sensitivity tests (same as Immerzeel et al. (2013)).**

| Sensitivity test | ΔT (°C) | ΔP (%) |
|---|---|---|
| Dry, cold | 1.5 | -3.2 |
| Dry, warm | 2.4 | -2.3 |
| Wet, cold | 1.3 | 12.4 |
| Wet, warm | 2.4 | 12.1 |

**Table 4: Confusion matrices for comparison of Landsat 8 snow maps and MOD10A2 snow maps with in situ snow observations.**

| | | MOD10A2 | | Landsat 8 | |
|---|---|---|---|---|---|
| | | Snow | No snow | Snow | No Snow |
| **In situ** | Snow | 83 | 31 | 20 | 3 |
| | No Snow | 75 | 438 | 18 | 106 |

**Table 5: Confusion matrices for comparison of in situ snow observations provided by snow depth and surface temperature observations with remotely sensed snow maps (MOD10A2 and Landsat 8 combined).**

| | | In situ snow depth | | In situ surface temperature | |
|---|---|---|---|---|---|
| | | Snow | No snow | Snow | No Snow |
| **Remotely** | Snow | 52 | 16 | 51 | 77 |
| **sensed** | No Snow | 17 | 80 | 17 | 464 |

**Table 6: Parameter value range prior to calibration and after calibration. The standard deviation of posterior parameter values is** 10 **based on the standard deviation of all members.**

| Parameter | Prior (min-max) | Posterior mean | Posterior std. |
|---|---|---|---|
| TT | -6 – 2 | -8.18 | 1.66 |
| $T_{lapse}$ | 0.9 – 1.10 | 1.10 | 0.01 |
| Precip | 0.6 – 1.4 | 1.31 | 0.02 |
| $C_6$ | 15 – 35 | 16.07 | 0.52 |

**Table 7: Classification accuracy of modelled snow extent based on pixel comparison with Landsat 8 snow maps. Calibrated accuracies are averaged over all members and the standard deviation represents the standard deviation in individual member accuracies (after calibration).**

| Date | Accuracy uncalibrated (%) | Accuracy calibrated (%) | Std. dev. accuracy (%) |
|------|---------------------------|-------------------------|------------------------|
| 02/11/13 | 80.96 | 84.41 | 0.12 |
| 18/11/13 | 78.43 | 79.15 | 0.11 |
| 04/12/13 | 77.41 | 77.10 | 0.05 |
| 20/12/13 | 54.97 | 60.38 | 0.08 |
| 05/01/14 | 63.46 | 67.07 | 0.07 |
| 20/01/14 | 74.30 | 81.33 | 0.04 |
| 06/02/14 | 65.55 | 73.24 | 0.05 |
| 10/03/14 | 84.94 | 89.67 | 0.05 |
| 26/03/14 | 87.03 | 86.90 | 0.04 |
| 11/04/14 | 80.29 | 82.92 | 0.05 |

