# Peer review of "Assimilation of snow cover and snow depth into a snow model to estimate snow water equivalent and snowmelt runoff in a Himalayan catchment"

_The Cryosphere, 2016_

## Referee Comment (RC1) · Anonymous Referee #1 · 30 Nov 2016

This paper presents a study focusing on the modelling of snow accumulation and melting in an Himalayan catchment and the response of this catchment under different climate scenarios in terms of snow water equivalent (SWE) and melt runoff. This study addressed an interesting topic in a region where snow storage is crucial for water supply. The authors use data assimilation (Ensemble Kalman Filter, EnKF) of ground-based and remotely-sensed snow data to determine optimal parameters values in their modelling system. These optimal parameters are then used in climate sensitivity tests.

My main comments about this study concern (i) the data assimilation method, especially the choice of variables to assimilate and the effects of these choices on final results and (ii) the limits of the climate sensitivity tests carried out with the optimized model. These questions need to be clarified prior to publication in TC. They are listed below (General comments) followed by more specific and technical comments.

**General Comments**

1) In the study, the EnKF is used to assimilate snow cover area per elevation band and snow depth at two locations. Four parameters are calibrated using the EnKF. My comments on this method concern (i) the choice and benefit of assimilating punctual snow depth measurements and (ii) the assimilation of MODIS snow cover.

The assimilation of punctual snow depth is associated with high uncertainties due to the very limited representativeness of punctual snow depth measurement in mountainous terrain (e.g. Grünewald and Lehning, 2015). For example, wind-induced snow transport can lead to erosion or accumulation of snow at the location of station. What would be the impact of such event when carrying out data assimilation with EnKF? Were the snow depth measurements assimilated in this paper impacted by such event? The benefit of directly assimilating snow depth measurement is hard to identify throughout the paper. It would be interesting to have results obtained when only snow cover data are assimilated. In the present version of the manuscript, the advantage of simultaneous assimilation of snow cover and depth is not clear enough. Results in Section 3.3.1 and 3.3.2 could be presented (i) without assimilation, (ii) with assimilation of snow cover only and finally (iii) with simultaneous assimilation of snow cover and depth.

The assimilation of MODIS snow cover requires an observation operator to convert SeNorge output into simulated snow cover extent. Are the authors using a simple threshold value of SWE or snow depth to determine the presence or the absence of snow? Or are they using depletion curves? MODIS snow cover are averaged per elevation band prior to assimilation. Can the author justify this choice? Indeed, averaging the information per elevation band reduce the information content brought by MODIS and remove the intra-band variability resulting from (i) the contrast between north-facing and south-facing slopes and (ii) the heterogeneous spatial distribution of precipitation.

2) The authors used the optimized version of their model to carry out climate sensitivity
tests. They use the delta method and applied changes in temperature and precipitation for different climate scenarios (Table 3). The authors do not discuss the uncertainties associated with this method. Such discussion is really relevant in a paper dealing with climate sensitivity. The delta method assumes constant changes in space and time for temperature and precipitation. How relevant is this assumption for this region? - Are the changes on temperature and precipitation expected to depend on the season? What are the expected effects for the hydrological cycle in this region? - The authors use the monthly precipitation pattern of Collier and Immerzeel (2015) to spatially distribute precipitation, both in present and future climate. The authors should discuss the validity of this assumption of constant monthly spatial pattern under future climate.

The study period (Jan. 2013 to Sep. 2014) should be compared to the present climatology of the catchment for temperature and precipitation. Is this period considered as cold or warm and wet or dry? Is it representative of the averaged current climate conditions in the Langtang catchment? The author apply the delta method to a short time period (from a climate perspective) and this short time period must be better characterized.

In section 3.5 at P 13 L1, L 13-14 and L 17-18, they authors discuss how the SWE and changes in SWE depend on elevation. This discussion is supported by Figures 7 and 8 that provide maps of SWE for the study period and change of SWE in the different climate sensitivity tests. I recommend the authors to provide complementary figures showing these variables as a function of elevation. It would help the reader to clearly identify the influence of elevation.

**Specific comments**

Introduction: the introduction is rather short and only presents earlier studies carried out in the Himalayan region. I recommend the authors to write more general paragraphs on (i) data assimilation of ground-based and remotely-sensed snow data in snowpack model and (ii) distributed snowpack modelling applied in mountainous region

to simulate the cryospheric and hydrological response of mountain catchments under present and future climate. They should present in this introduction how techniques developed in other mountainous regions could be applied to an Himalayan catchment.

P 4 L 16-17: the description of the location of the snow depth measurements is confusing. Are the 4 sites measuring snow depth located along the 2 transects? Figure 1 suggests that this is not the case. The authors should clarify this point.

P 4 L 28: which uncertainties are taken into account with the correction factor *precip*? Does it include: - uncertainties in solid precipitation measurements at the station due to wind undercatch? - spatial and temporal representativeness across the catchment of the precipitation measured at the station?

P 6 L 3-4: please mention that in Brock et al. (2000) the snow albedo remains constant when the maximum air temperature is below 0 °C.

P 6 L 22: the sentence "Separate transport ... this study" should be reformulated. It suggests than when wet snow avalanches occur the ice and liquid phases are transported separately. This is not the case in the nature. It seems that the authors mentions this point only because seNorge treats separately the solid and the liquid phase in the snowpack.

P 7 L 8: the runs used for the sensitivity analysis are not clearly described. For each run, are the authors using the model to simulate the evolution of snow cover and SWE over the whole study period (January 2013- September 2014) and the whole catchment? Or are they using different time period and sub-domains?

P 7 L 10: how are computed the mean snow cover extent and snow depth? Are they averaged over the whole period and the whole domain? This point is similar to my previous point regarding the characteristics of the simulations used in the sensitivity analysis.

P 7 L 10 (and in the rest of the paper): the author should precise how they compute

the snow cover extent from the output of seNorge. Cf my general comments about the observation operator.

P 8 L 22-23: how is modified the maximum air temperature in the climate sensitivity tests?

P 10 L 19-25: this paragraph should also discuss model results in the elevations zones above 5000 m. For example, could the author discuss the differences between summer 2013 and 2014 in terms of snow extent in the elevation zones 5000-5000 m and >5500 m? What can explain the underestimation of SCE in these zones for summer 2013 whereas better results are achieved in summer 2014?

P 10 L 29: differences in classification accuracy with and without calibration are hard to identify on Figure 4. A map of differences of classification accuracy could help the reader to better identity the regions where large differences are found between the two simulations.

P 10 L 30-31: the authors associate the low classification accuracies in the northern part of the catchment with model errors due the avalanching parametrization. However, it seems that this difference can also arise from errors in the meteorological forcing used to drive seNorge. For example: (i) errors in precipitation phase and amount, (ii) errors in the spatial distribution of precipitation. Indeed, the spatial distribution of precipitation is based on monthly precipitation patterns derived from Collier and Immerzeel (2015). For a given precipitation event, the spatial distribution of precipitation can vary from the monthly pattern from Collier and Immerzeel (2015) and strongly affect the snow cover. Please add a discussion about the different potential sources of error.

P 11 L 23-24: please consider reformulating the last sentence of this paragraph. Indeed, the improvement for Kyangjin in 2014 is not really clear.

P 11 L 25: the authors point out the lack of independent stations for the evaluation

of snow depth and SWE. Are glacier mass balance data available for a glacier in this catchment to bring complementary values for evaluation? For example, winter mass balance data can provide interesting evaluation on the cumulated precipitation during the winter.

P 11 L32: the absence of underestimation or overestimation concerns snow depth and not SWE.

P 12 L 5-30, Section 3.4: This section does not contain new and original results and only presents the effect of well-established parametrizations introduced in seNorge to improve the snowpack dynamics without comparison with measurements. I recommend the authors to remove the discussion concerning the snow compaction and the snow albedo since it does not bring additional value to their paper. Concerning the avalanche parametrization, the discussion at lines 7-10 (P 12) suggests that avalanching strongly affects the simulation results. It would be really interesting if the authors could illustrate how the avalanching parametrization improves the representation of the snow depth distribution in the model. Figure 7 shows that, in the simulations, snow accumulates at the bottom of the steep slopes of the catchment. Are these zones of additional snow accumulation identified on the LandSat images at 30-m resolution? Such discussion on avalanche processes and a comparison with remotely-sensed observation would substantially improve the quality of this section on snow processes. Otherwise, I recommend to remove this section from the paper.

**Technical comments**

**Text**

P 16 L 25: modify the reference to Immerzeel et al. (2014)

**References** (not included in the submitted manuscript)

Grünewald, T., Lehning, M. (2015). Are flat‐field snow depth measurements representative? A comparison of selected index sites with areal snow depth measurements

at the small catchment scale. Hydrological Processes, 29(7), 1717-1728.

---

## Referee Comment (RC2) · H. Wulf (Referee) · 20 Feb 2017

Dear Editor, author and co-authors.

This paper presents an illuminating analysis in the spatial (and temporal) distribution of SWE in the Langtang Valley, Nepal, and the findings are useful to anyone investigating the hydroclimatic phenomena and variability from the Hindu Kush to the Eastern Himalaya. Unique to this study is the use of actual field observation (a rarity in the Himalayas) and their incorporation in a modeling approach fo SWE. The explication could be improved, and I have marked up the manuscript. But although my comments are extensive, they are straightforward, so I do not feel I need to see the paper again. I look forward to reading the published version.

Kind regards, Hendrik Wulf

[Figure]

Please also note the supplement to this comment:
http://www.the-cryosphere-discuss.net/tc-2016-216/tc-2016-216-RC2-supplement.pdf

**Supplement:**

[revised manuscript text omitted]

---

## Referee Comment (RC3) · Anonymous Referee #3 · 27 Feb 2017

General: The paper presents an interesting analysis of current and future snow dynamics for the Langtang catchment in the Himalayas (Nepal). The paper is well-written and follows a clear line of argumentation. The approach of including as much as possible local and satellite data has a lot of merit. At the same time, I have major concerns about the methodology as detailed in the following. In my opinion, climate change scenario calculations based on simple (parameterized) snow models are unreliable as they necessarily present an extrapolation beyond the state for which the models have been calibrated. The problem with such simple snow models has been exemplarily shown by Magnusson et al. (2011). In this publication, a physics-based model and a model similar to seNorge are shown to produce similar results for a current climate but very diverging results for climate change scenarios. The data assimilation via Kalman filtering potentially makes the modelling in the presented paper even more vulnerable

to extrapolation than when using robust standard parameters. This is a major objection I have towards the methodology. What is aggravating the problem described above is that the paper appears to completely ignore a large body of literature, which is based on physics-based snow modelling of climate change impacts. This has already been pointed out by RC1. I do not want to necessarily suggest that a paper based on temperature index modelling of future climate needs to be rejected in all cases. But if such an analysis is retained it needs to show a very careful assessment of potential errors through extrapolation and a discussion and comparison with results obtained with physics-based models. Computational restrictions do no longer prevent physics-based models to be applied to larger areas and for significant climate change studies. A recent example is Marty et al. (2017), which has just appeared in TC and which is a good starting point for the authors to find additional studies, which they need to discuss in context of their analysis. Interestingly, the results of the latter study (for the Alps) qualitatively agree with what the authors find for Langtang and this is a good sign. But this also means that the results are qualitatively not new and quantitatively highly uncertain for the argument presented above. This is my major point about the paper and I otherwise agree with the points raised by RC1. In general, presentation, figures and form of the paper are already at a very advanced state and almost without problems.

References:

Magnusson, J., Farinotti, D., Jonas, T. and Bavay, M. (2011), Quantitative evaluation of different hydrological modelling approaches in a partly glacierized Swiss watershed. Hydrol. Process., 25: 2071–2084. doi:10.1002/hyp.7958 Marty, C., Schlögl, S., Bavay, M., and Lehning, M.: How much can we save? Impact of different emission scenarios on future snow cover in the Alps, The Cryosphere, 11, 517-529, doi:10.5194/tc-11-517-2017, 2017.

---

## Author Comment (AC1) · 17 Mar 2017

**Reply to Anonymous Referee #1**

*We thank Anonymous Referee #1 for the thorough review. We give our reply (in italic) to the referee comments/ suggestions below.*

This paper presents a study focusing on the modelling of snow accumulation and melting in an Himalayan catchment and the response of this catchment under different climate scenarios in terms of snow water equivalent (SWE) and melt runoff. This study addressed an interesting topic in a region where snow storage is crucial for water supply. The authors use data assimilation (Ensemble Kalman Filter, EnKF) of groundbased and remotely-sensed snow data to determine optimal parameters values in their modelling system. These optimal parameters are then used in climate sensitivity tests. My main comments about this study concern (i) the data assimilation method, especially the choice of variables to assimilate and the effects of these choices on final results and (ii) the limits of the climate sensitivity tests carried out with the optimized model. These questions need to be clarified prior to publication in TC. They are listed below (General comments) followed by more specific and technical comments.

**General Comments**
1) In the study, the EnKF is used to assimilate snow cover area per elevation band and snow depth at two locations. Four parameters are calibrated using the EnKF. My comments on this method concern (i) the choice and benefit of assimilating punctual snow depth measurements and (ii) the assimilation of MODIS snow cover. The assimilation of punctual snow depth is associated with high uncertainties due to the very limited representativeness of punctual snow depth measurement in mountainous terrain (e.g. Grünewald and Lehning, 2015).

*We are aware of the high uncertainties related to the limited representativeness of punctual snow depth observations in complex terrain due to local influence of snow drift. We will add the reference, provided by the reviewer, to the revised manuscript for completeness. A key advantage of the EnKF is that it takes into account the uncertainty in the assimilated observations. Several observation uncertainties were tested (variance of 1cm, 16cm and 25cm) on how it influenced the posterior parameter distribution, prior to choosing the final observation uncertainty that is presented in this manuscript (variance of 25 cm). See the next reply for explanation on why we believe that a variance of 25 cm is representative for the uncertainty of punctual snow depth measurements in this case.*
*In addition we want to emphasize that the assimilation of snow depth observations is only used to calibrate the compaction parameter $C_6$ and not for calibration of the other three parameters.*

For example, wind-induced snow transport can lead to erosion or accumulation of snow at the location of station. What would be the impact of such event when carrying out data assimilation with EnKF? Were the snow depth measurements assimilated in this paper impacted by such event?

*Wind-induced snow transport is not included in the snow model presented in this study. If the snow depth measurements are affected by wind-induced snow transport there will definitely be a discrepancy between simulated and measured snow depth that cannot be correctly explained by the model. Using the EnKF for assimilation of measured snow depth to obtain optimal parameter values allows accounting for the uncertainty related to point snow depth measurements. Using a too small measurement uncertainty in the EnKF would result in model parameter values that are too strongly driven by the assimilated snow depth measurements, resulting in implausible values of the model parameters. We checked to which degree the parameter values were driven too strongly by the snow depth measurements as cause of a discrepancy between simulated and measured snow depth. This discrepancy*

*could (amongst other snow processes) result from wind-induced snow transport. It is difficult to assess whether the station data is influenced by this process based on only measurements of the snow depth. However, results from using different measurement uncertainties showed that a variance of 25 cm (that is used in this study) resulted a mild forcing of the model parameters and plausible values. Therefore, a variance of 25 cm for the measurement uncertainty was assumed to be representative for the uncertainty of punctual snow depth measurements in this case.*

The benefit of directly assimilating snow depth measurement is hard to identify throughout the paper. It would be interesting to have results obtained when only snow cover data are assimilated. In the present version of the manuscript, the advantage of simultaneous assimilation of snow cover and depth is not clear enough. Results in Section 3.3.1 and 3.3.2 could be presented (i) without assimilation, (ii) with assimilation of snow cover only and finally (iii) with simultaneous assimilation of snow cover and depth.

*The data assimilation is performed in a two-step approach (p8 l14-18). TT, Tlapse and precip were optimized by assimilation of snow extent, whereas $C_6$ was optimized by assimilation of snow depth. In section 3.3.1 distinction can only be made between the steps 'without assimilation' and 'with assimilation' of snow cover, respectively uncalibrated and calibrated in Figures 3 and 4 because $C_6$ does not influence the snow extent as it is an parameter that converts SWE in snow depth.*
*The simulated snow depth without assimilation ('uncalibrated') and after assimilation of both snow depth and snow extent ('calibrated') is already given in Figure 5. The simulated snow depth after assimilation of snow cover only will be included to this figure to show the advantage of assimilation of both snow extent and snow depth (see* Figure 1 *below).*

[Figure]

**Figure 1 Observed snow depth and modelled snow depth before calibration, after assimilation of snow extent, and after assimilation of both snow depth and snow extent (ensemble mean) at four locations. The RMSE (mm) is given for the fit between modelled (before calibration, after assimilation of snow extent, and after assimilation of both snow depth and snow extent) and observed snow depth.**

The assimilation of MODIS snow cover requires an observation operator to convert SeNorge output into simulated snow cover extent. Are the authors using a simple threshold value of SWE or snow depth to determine the presence or the absence of snow? Or are they using depletion curves?

*The presence of snow in the model is based on a threshold value of 1 mm swe. This will be added to the manuscript.*

MODIS snow cover are averaged per elevation band prior to assimilation. Can the author justify this choice? Indeed, averaging the information per elevation band reduce the information content brought

by MODIS and remove the intra-band variability resulting from (i) the contrast between north-facing and south-facing slopes and (ii) the heterogeneous spatial distribution of precipitation.

*The assimilation of snow cover would preferably be performed on a pixel to pixel basis to maintain all information on the spatial distribution of snow cover. However, the EnKF can only be used for continuous values and not for binary values (i.e. snow cover present or not). Therefore it is required to assimilate snow extent (continuous value) into the model. Rather than using the total snow extent for the entire catchment, we chose elevation bands to include more information on the spatial variability of snow cover. Elevation bands were chosen to capture the snow elevation line transition and therefore capture melt dynamics and spatial distribution of precipitation.*

2) The authors used the optimized version of their model to carry out climate sensitivity tests. They use the delta method and applied changes in temperature and precipitation for different climate scenarios (Table 3). The authors do not discuss the uncertainties associated with this method. Such discussion is really relevant in a paper dealing with climate sensitivity. The delta method assumes constant changes in space and time for temperature and precipitation. How relevant is this assumption for this region? - Are the changes on temperature and precipitation expected to depend on the season? What are the expected effects for the hydrological cycle in this region? - The authors use the monthly precipitation pattern of Collier and Immerzeel (2015) to spatially distribute precipitation, both in present and future climate. The authors should discuss the validity of this assumption of constant monthly spatial pattern under future climate.

*The scope of this study is not to run long-term climate change runs as indeed the study period is too short and might not be representative. This short study period simply does not allow a full-fledged study on climate change scenarios. The focus is rather on the climate sensitivity of the SWE in this study area as a result of changes in temperature and precipitation (using the simple delta method). The study shows the combined effect of changes in air temperature and precipitation. We aim to show patterns that could potentially occur in future under changes in temperature and precipitation. The four RCP4.5 scenarios from Immerzeel et al. (2013) were mainly used to have realistic values for changes in precipitation and temperature.*

*The spatial distribution of precipitation was kept constant for simplicity. Changes in (spatial distribution of) precipitation are difficult to predict and simulate. In this study we aim to show the importance of knowing the spatial distribution of precipitation in future as we show that increased melt due to increased air temperature can be compensated by an increase in precipitation at high elevation.*

*So, by no means do we intend to present a climate change impact study, but merely a sensitivity study. In the revised manuscript we will make this clearer. The core of the study is to show how assimilation of snow depth and remotely sensed snow cover in a snow model can lead to a better understanding and quantification of snow water equivalent and snowmelt runoff in an inaccessible, data scarce environment. In particular the title may have given the reviewer the wrong impression about the focus of our study and we will modify the title: 'Assimilation of snow cover and snow depth in a snow model to estimate snow water equivalent and snowmelt runoff in a Himalayan catchment'.*

The study period (Jan. 2013 to Sep. 2014) should be compared to the present climatology of the catchment for temperature and precipitation. Is this period considered as cold or warm and wet or dry? Is it representative of the averaged current climate conditions in the Langtang catchment? The author

apply the delta method to a short time period (from a climate perspective) and this short time period must be better characterized.

*The study period will be compared to climatology of the catchment. An additional figure (see Figure 2 below) will be added comparing the study period to the 1988-2009 climatology. However, we want to emphasize again that we do not intend to present a climate change impact study, but merely a sensitivity study.*

[Figure]

**Figure 2 Comparison of maximum temperature (T$_{max}$) and cumulative monthly precipitation (P) for the study period and the 1988-2009 time series (based on measurements in Kyangjin). The average yearly cumulative precipitation is 853mm and 663mm for the study period and the 1988-2009 time series respectively.**

In section 3.5 at P 13 L1, L 13-14 and L 17-18, they authors discuss how the SWE and changes in SWE depend on elevation. This discussion is supported by Figures 7 and 8 that provide maps of SWE for the study period and change of SWE in the different climate sensitivity tests. I recommend the authors to provide complementary figures showing these variables as a function of elevation. It would help the reader to clearly identify the influence of elevation.

*An additional figure will be added with boxplots of SWE per elevation zone for the reference run and the four climate sensitivity tests (see* Figure 3 *below).*

[Figure]

**Figure 3 Boxplots of SWE per elevation zone averaged over the simulation period and all ensemble members for the reference run and the four climate sensitivity tests.**

**Specific comments**

Introduction: the introduction is rather short and only presents earlier studies carried out in the Himalayan region. I recommend the authors to write more general paragraphs on (i) data assimilation of ground-based and remotely-sensed snow data in snowpack model and (ii) distributed snowpack modelling applied in mountainous region to simulate the cryospheric and hydrological response of mountain catchments under present and future climate. They should present in this introduction how techniques developed in other mountainous regions could be applied to an Himalayan catchment.

*A more extensive introduction will be given, including the points given above.*

P 4 L 16-17: the description of the location of the snow depth measurements is confusing. Are the 4 sites measuring snow depth located along the 2 transects? Figure 1 suggests that this is not the case. The authors should clarify this point.

*There are 2 transects of surface temperature measurements. Only two out of four snow depth measurements are located along the 2 transects. This will be adjusted.*

P 4 L 28: which uncertainties are taken into account with the correction factor precip? Does it include: - uncertainties in solid precipitation measurements at the station due to wind undercatch? - spatial and temporal representativeness across the catchment of the precipitation measured at the station?

*Correction factor precip accounts for the uncertainties related to undercatch. This factor does not account for the uncertainties in the representativeness of the chosen monthly spatial distribution of precipitation. We believe that quantifying the uncertainties in spatial distribution is beyond the scope of this study. The fact that we use the monthly spatial precipitation distribution based on a high resolution weather model based Collier and Immerzeel (2015) is already a great improvement over previous studies where fixed lapse rates were used for the entire catchment to regionalize precipitation based on observations of a single station.*

P 6 L 3-4: please mention that in Brock et al. (2000) the snow albedo remains constant when the maximum air temperature is below 0 _C.

*This will be added.*

P 6 L 22: the sentence "Separate transport ... this study" should be reformulated. It suggests than when wet snow avalanches occur the ice and liquid phases are transported separately. This is not the case in the nature. It seems that the authors mentions this point only because seNorge treats separately the solid and the liquid phase in the snowpack.

*This was indeed mentioned because seNorge separately treats the solid and liquid phase in the snowpack. The sentence will be reformulated.*

P 7 L 8: the runs used for the sensitivity analysis are not clearly described. For each run, are the authors using the model to simulate the evolution of snow cover and SWE over the whole study period (January 2013- September 2014) and the whole catchment? Or are they using different time period and sub-domains?

*The sensitivity runs simulate the evolution of snow cover and SWE over the whole study period and the whole catchment. This will be better described.*

P 7 L 10: how are computed the mean snow cover extent and snow depth? Are they averaged over the whole period and the whole domain? This point is similar to my previous point regarding the characteristics of the simulations used in the sensitivity analysis.

*They are indeed averaged over the whole study area and study period. The description will be improved.*

P 7 L 10 (and in the rest of the paper): the author should precise how they compute the snow cover extent from the output of seNorge. Cf my general comments about the observation operator.

*See the answer on the general comment.*

P 8 L 22-23: how is modified the maximum air temperature in the climate sensitivity tests?

*This is a good point. In the previous version of the manuscript we have only perturbed the mean temperature forcing, but for the revised manuscript we have now imposed the same delta change on the maximum temperatures. The impact is however minimal and only for the dry warm case, a difference of a few percent in SWE is simulated compared to the previous version. This is because the maximum temperature is only used in the albedo decay algorithm, so when Tmax is higher, then the albedo decay will start a bit earlier, however the impact is minimal in these sensitivity tests. Figures 8 and 9 will be updated in the revised manuscript with the climate sensitivity tests including perturbation of Tmax.*

P 10 L 19-25: this paragraph should also discuss model results in the elevations zones above 5000 m. For example, could the author discuss the differences between summer 2013 and 2014 in terms of snow extent in the elevation zones 5000-5000 m and >5500 m? What can explain the underestimation of SCE in these zones for summer 2013 whereas better results are achieved in summer 2014?

*We will add a paragraph to the results and discussion section, where we discuss the potential reasons for those high altitude differences.*

P 10 L 29: differences in classification accuracy with and without calibration are hard to identify on Figure 4. A map of differences of classification accuracy could help the reader to better identity the regions where large differences are found between the two simulations.

*Before calibration the snow model already shows high performance in simulating snow cover (Figure 4a). After calibration there is only a modest improvement in accuracy in most regions, whereas there is a slight decrease in performance in few other regions. The small changes in performance are therefore indeed hard to identify. The only pronounced improvement is in the lower area on the northern slope and this improvement also shows from Figure 3. A difference map was tested, but resulted in rather chaotic patterns of small increases and decreases in accuracy in steep terrain and therefore does not help visualization and interpretation. Given the little information content, the fact that elevation dependent improvements are already shown in Figure 3 and since we already add a new figure (boxplots) we have decided not to include the proposed figure in the revised manuscript.*

P 10 L 30-31: the authors associate the low classification accuracies in the northern part of the catchment with model errors due the avalanching parametrization. However, it seems that this difference can also arise from errors in the meteorological forcing used to drive seNorge. For example: (i) errors in precipitation phase and amount, (ii) errors in the spatial distribution of precipitation. Indeed, the spatial distribution of precipitation is based on monthly precipitation patterns derived from Collier and Immerzeel (2015). For a given precipitation event, the spatial distribution of precipitation can vary from the monthly pattern from Collier and Immerzeel (2015) and strongly affect the snow cover. Please add a discussion about the different potential sources of error.

*This is correct and a more complete description of reasons for low classification errors will be added to the discussion.*

P 11 L 23-24: please consider reformulating the last sentence of this paragraph. Indeed, the improvement for Kyangjin in 2014 is not really clear.

*This will be reformulated.*

P 11 L 25: the authors point out the lack of independent stations for the evaluation of snow depth and SWE. Are glacier mass balance data available for a glacier in this catchment to bring complementary values for evaluation? For example, winter mass balance data can provide interesting evaluation on the cumulated precipitation during the winter.

*The yearly mass balance is available for Yala glacier, which is a clean-ice glacier positioned in the study area. However, the yearly mass balance is negative. As only snowmelt is simulated, and no glacier melt, it is impossible to simulate a negative mass balance with the snow model. Therefore no comparison was made between output of the snow model and available mass balance data.*

*Though, an extra data set of snow depth measurements will be used (Yala BC; see* Figure 1 *above) to improve the validation of the simulated snow depth. This will be added to the revised manuscript.*

P 11 L32: the absence of underestimation or overestimation concerns snow depth and not SWE.

*This will be adapted.*

P 12 L 5-30, Section 3.4: This section does not contain new and original results and only presents the effect of well-established parametrizations introduced in seNorge to improve the snowpack dynamics without comparison with measurements. I recommend the authors to remove the discussion concerning the snow compaction and the snow albedo since it does not bring additional value to their paper.

*This section will be removed from the manuscript.*

Concerning the avalanche parametrization, the discussion at lines 7-10 (P 12) suggests that avalanching strongly affects the simulation results. It would be really interesting if the authors could illustrate how the avalanching parametrization improves the representation of the snow depth distribution in the model. Figure 7 shows that, in the simulations, snow accumulates at the bottom of the steep slopes of the catchment. Are these zones of additional snow accumulation identified on the LandSat images at 30-m resolution? Such discussion on avalanche processes and a comparison with remotely-sensed observation would substantially improve the quality of this section on snow processes. Otherwise, I recommend to remove this section from the paper.

*The snow accumulation zones are not clearly visible in Landsat 8 imagery. This is caused by i) the shadow in the steep areas, ii) a too small extent of the accumulation zones, and iii) potentially a wrong timing of the acquisition date. Therefore a comparison of simulated and remotely sensed snow cover in the deposition zones is impossible. Based on extensive field experience we know that avalanches regularly occur at these steep walls and that snow is deposited downslope these steep walls. The walls are too steep to have a substantial snow depth. Including avalanching in the model improves the snow redistribution as there can be no unrealistic accumulation of snow in these steep zones. Although, there is no possibility to support this with Landsat 8 imagery, we respectfully disagree with the recommendation of removing this section, as we believe it is important and improves the distribution of snow in the model.*

**Technical comments**
**Text**

P 16 L 25: modify the reference to Immerzeel et al. (2014)

*This will be modified.*

**References** (not included in the submitted manuscript)
Grünewald, T., Lehning, M. (2015). Are flat-field snow depth measurements representative? A comparison of selected index sites with areal snow depth measurements

*This will be included.*

---

## Author Comment (AC2) · 17 Mar 2017

**Reply to Hendrik Wulf Referee #2**

*We thank Hendrik Wulf for his extensive comments and positive feedback. All the suggested typographical revisions will be adjusted for the revised manuscript. Below we summarize the comments given by the reviewer and our replies (italic)*

Dear Editor, author and co-authors.
This paper presents an illuminating analysis in the spatial (and temporal) distribution of SWE in the Langtang Valley, Nepal, and the findings are useful to anyone investigating the hydroclimatic phenomena and variability from the Hindu Kush to the Eastern Himalaya. Unique to this study is the use of actual field observation (a rarity in the Himalayas) and their incorporation in a modeling approach for SWE. The explication could be improved, and I have marked up the manuscript. But although my comments are extensive, they are straightforward, so I do not feel I need to see the paper again.
I look forward to reading the published version.
Kind regards, Hendrik Wulf

p1 l20 Why only in the Himalayas? Isn't this of general interest in complex snowy terrain?

*It is a new approach for the Himalayas and therefore of main interest for the Himalayas. However, it is indeed of general interest for complex terrain. We will stress this in the revised manuscript.*

p1 l22 Why do you assume an increase in precipitation at high elevation in the future? Is that true for all your four scenarios or which one do you focus on here? I find this a little bit confusion. A suggestion. Pick the most likely out of your four scenarios and provide the impact of the change with some numbers. So, what is the change in temperature and what would be the impact on SWE.

*Immerzeel et al. (2013) analysed all available CMIP5 simulations for the emission scenario RCP4.5 for the Langtang catchment. They selected the four extremes from the RCP4.5 ensemble members ranging from dry to wet and from cold to warm. We used their projected changes in precipitation and temperature (Table 3). Their projected changes show both a decrease and increase in precipitation. We want to emphasize that we do not intend to perform a climate change impact study but merely a sensitivity study (see also the reply to reviewer #3). We describe the patterns that result from the climate sensitivity tests (including increase and decrease in precipitation). There is not per definition a climate sensitivity test that is most likely. Therefore a description is given of the most interesting results from the sensitivity tests, i.e. patterns that occur as result of changes in temperature and precipitation.*

p1 l25 Some kind of closing sentence is missing with regard to the opening. For example, snow as important water storage in the Langtang Valley is projected to decrease by X% assuming a temperature increase of X°C, which has implications on ...

*No numbers are presented here to prevent the thought that we actually performed a climate impact study. We rather performed climate sensitivity tests and we intend to mainly describe qualitative results.*

p2 l11 "No information" is not quite correct. Using the "inverse melt" approach by Molotch et al. (2009) you can gain information on SWE. I used this simple approach in the western Himalaa and it worked quite well (Wulf et al. 2016)

MolotchNP,NorteD.ReconstructingsnowwaterequivalentintheRioGrandeheadwatersusingremotelysense
dsnowcoverdataandaspatiallydis-tributedsnowmeltmodel.HydrolProcess2009;1089:1076–
89.http://dx.doi.org/10.1002/hyp.7206.

*It will be changed into 'limited information'. The given references will be added. We appreciate the suggestion.*

p2 l12 Do you refer to currently recorded data or available data? I am aware of continuos data records in the Indian Himalaya, which are not publicly available. Further efforts existed in the central Himalaya. There are publications on snowfall and SWE in the Himalaya. See Putkonen et al.(2004) and Wulf et al. (2016).

Putkonen, J.K.
Continuous snow and rain data at 500 to 4400 m altitude near Annapurna, Nepal, 1999-2001
(2004) Arctic, Antarctic, and Alpine Research, 36 (2), pp. 244-248. Cited 38 times.
https://www.scopus.com/inward/record.uri?eid=2-s2.0-
9944234078&partnerID=40&md5=c770cdf50445a4762d9b9757e46f56de

Wulf, H., Bookhagen, B., Scherler, D.
Differentiating between rain, snow, and glacier contributions to river discharge in the western Himalaya using remote-sensing data and distributed hydrological modeling
(2016) Advances in Water Resources, 88, pp. 152-169. Cited 1 time.
https://www.scopus.com/inward/record.uri?eid=2-s2.0-
84954410375&doi=10.1016%2fj.advwatres.2015.12.004&partnerID=40&md5=a48d65ad590068da9a36
3c979cd7b15b

*This sentence will be rephrased: 'Currently there is limited information of SWE for the Himalayas'.*
*We were referring to available data sets. Thank you for making us aware of this interesting literature. The references will be added to the revised manuscript.*

p2 l30 You highlight the differences between low-elevation and high-elevation precipitation at the southern slopes. Is there also a north south gradient in precipitation?
See the work of Ana Barros.

*We will rephrase this as follows:*
*There is a strong interaction between the orography and precipitation patterns. During the monsoon, at the synoptic scale, there is a decreasing trend from south to north during the monsoon, but at smaller scales there are more local orographic effects associated to the aspect of the main valley ridges (Barros et al, 2004) that determine the precipitation distribution. During the monsoon precipitation mainly accumulates at the south western slopes near the catchment outlet at low elevation. Winter westerly events can also provide significant snowfall. Snow cover has strong seasonality with extensive, but sometimes erratic, winter snow cover and retreat of the snowline to higher elevations during spring and summer and less snow cover. During the winter precipitation mainly accumulates along high-elevation southern-eastern slopes (Collier and Immerzeel, 2015).*

p3 l25 I would recommend to use the more general formulation: NDSI (Green-SWIR1) / (Green+SWIR1). This way you could reuse the equation for L8, too.

*The formula for the NDSI will be changed into a more general formula using green and SWIR.*

p4 l2 How many cloud free scenes out of how many total scenes did you use in the end? How well did they cover the snow melt period?

*10 out of 34 available Landsat 8 images were used for validation of the snow model. The coverage can be seen in Figure 3.*

p4 l14 What did you use the surface temperature for?

*We used the surface temperature measurements for distinguishing between snow covered and no-snow covered periods at a point-scale. These point measurements are used to validate the remotely sensed snow cover. This is described in the results and discussion section.*

p4 l16 Surface or air temperature?

*Surface temperature. This will be revised.*

p4 l25 What are the lapse rate values? How good do they describe the variation between the temperature stations? Do these values vary during the year?

*The temperature lapse rates agree with values presented in the study of Immerzeel et al. (2014). There is seasonal variation in the lapse rates similar to Immerzeel et al. (2014).*

p4 l29 How uncertain are these temperature measurements?

*We do not refer to the uncertainty in the temperature measurements, but to the uncertainty of the derived temperature lapse rate. We will clarify this in the revised manuscript.*

p5 l9 How did you incorporate the 500m MODIS pixel in your model? Simple upscaling?

*The 500m MODIS pixels were resampled to 100m to fit the spatial resolution of the model.*

p6 l10 Do you have any idea by which degree your snow depth measurements are affected by snow redistribution?

*The snow depth measurements are not influenced by avalanching, though there might be some influence from wind transport. However, when assimilating the snow depth measurements, an uncertainty was added to the snow depth measurement to account for the uncertainty that results from wind-induced snow deposition and erosion. See also the reply to reviewer 1.*

p7 l29 I assumed you use a distributed modelling approach. Do I rightly assume that snowmelt is generated per elevation band not per model pixel? Please clarify.

*Snowmelt is generated per pixel. This will be clarified.*

p7 l30 Landsat snow cover data is not used in your model only the validation?

*Yes, it is used as an independent validation of the simulated snow cover.*

p7 l31 Why did you not choose equal area breakpoints to ease the direct comparison?

*We want to characterize the snow cover per elevation zone. Equal area breakpoints resulted in unevenly distributed elevation zones given the catchment's hypsometry. Therefore, unequal area breakpoints were chosen to have approximate equal elevation intervals.*

p8 l24 What does this mean?

*Immerzeel et al. (2013) analysed all available CMIP5 simulations for the emission scenario RCP4.5 and extracted precipitation and temperature trends. They selected 4 models that ranged from dry to wet and from cold to warm for the Langtang catchment. The changes in in temperature (°C) and precipitation (%) are extracted from the 4 models. The projected annual change is for 2021-2050 relative to 1961-1990. This will be clarified in the manuscript.*

p8 l25 What about the dry to dryer scenario is the summer monsoon and/or westerlies weaken?

*It is assumed that both occur. However, precipitation is much more substantial during monsoon and therefore the dry to dryer climate sensitivity tests mainly influence the monsoon precipitation.*

p8 l26 I recommend to clearly distinguish between your results and the discussion. This is common scientific practice.

*We believe that combining these sections improves the readability in our case. It is nowadays also not uncommon to combine the results and discussion, so we propose to keep it as it is..*

p9 l12 The other studies compared MOD10A1 data. The simplification of MOD10A2 surely introduces additional errors. Larger uncertainties also stem from large viewing angles, which can increase the observation area by a factor of 10 for MODIS. See Dozier et al. 2008.

*Both notes will be added to the manuscript.*

p9 l17 I would assume that the snow observation on the ground differs, too. I honestly doubt that relief introduces such a big error if the satellite data is geocoded correctly. USGS does a good job here for their TOA products.

*In situ snow observations indeed result in additional uncertainty. This is already described in the manuscript (p9 l17-22). The relief is causing high spatial variability in snow cover. It is believed that the spatial resolution of MOD10A2 snow cover does not capture this spatial variability. We do not refer to correct geocoding. This will be clarified.*

p10 l6 Any values (before and after calibration) would be much appreciated.

*These values can be found in Table 6. We already refer to this table in the manuscript (p10 l2).*

p15 l4 Do you mean: "increased melt due to higher temperatures"?

*Yes, this will be revised.*

p18 l14 Nice figure. However, the map is missing coordinated and an inset to locate it in the Himalayas.

*The figure will be revised so it addresses these comments (See* Figure 1 *below).*

[Figure]

**Figure 1 Study area with the locations of the in situ observations. Langtang and Langshisha refer to two main glaciers in upper Langtang Valley.**

p19 l1 Which are these locations and what is their respective elevation? Why is the surface temperature above zero for some snow cover periods? How certain are you about the snow free periods during sprng the upper example (Yala 5)? Are late snowfall events more common in spring as compared to the winter period?

*The locations and elevation are given in Table 1. The table number will be added to the figure description.*
*The above zero temperature results from the uncertainty of the surface temperature measurements. The short snow free periods during spring for Yala 5 are associated with the uncertainty of the temperature measurements. During winter precipitation events are rare, whereas during spring late snowfall events are common.*

In Figure 3 there seems to be quite a mismatch between Landsat and MODIS. How do you explain this difference?

*The coarse resolution of MODIS does not allow observing the high spatial variability in snow cover. MODIS snow cover shows full snow cover, whereas Landsat shows higher spatial variability (also some no-snow cover) at higher elevations. This results in a mismatch between MODIS and Landsat.*
*In addition the Landsat 8 derived snow maps are influenced by shading. Shaded snow covered area is erroneously mapped as no-snow areas and therefore also results in underestimation of snow cover by landsat compared to MODIS.*

p21 l5 Could you indicate these locations in your Fig. 1? I would opt for dashed and solid lines to improve the distinction in b/w printouts.

*These locations are already indicated in Table 1 and Figure 1.*
*The figure will be updated and will contain more lines (see reply to reviewer 1). Therefore we believe that dashed and solid lines will not enhance distinguishing between different lines.*

Do the values in Figure 7 match the glacier snow accumulation rate at high elevations? What is the maximum SWE value you got?

*We do not know the snow accumulation rate at high elevations as there is only yearly mass balance data available that also includes ablation.*
*The maximum SWE will be given in an additional figure showing box plots of SWE values per elevation zone (see reply to reviewer 1).*

In Figure 8 you have (white) space enough to write the different scenarios directly into the figure (as a subfigure headline. Please also state here what wet, dry, cold and warm refer to.

*This will be adjusted. Reference will be made to Table 3 to clarify the different climate sensitivity tests mentioned in this figure.*

p24 l1 How was runoff measured? Which method? Does melt also include glacier melt? If not, it is better to refer to snow melt.

*Runoff was not reliably measured in the field and is therefore not included in this study. In the model it is assumed that all runoff, for each pixel, collects at the catchment outlet for each time step without applying routing.*
*Melt includes only snowmelt, although it includes snowmelt on glaciers. The legend will be adapted to snowmelt.*

*References:*
*Barros, A. P., Kim, G., Williams, E. and Nesbitt, S. W.: Probing orographic controls in the Himalayas during the monsoon using satellite imagery, Nat. Hazards Earth Syst. Sci., 4(1), 29–51, doi:10.5194/nhess-4-29-2004, 2004.*

---

## Author Comment (AC3) · 17 Mar 2017

**Reply to Anonymous Referee #3**

We thank Anonymous Referee #3 for the thorough review. We provide our reply to the referee suggestion/comments below (in italics)

General: The paper presents an interesting analysis of current and future snow dynamics for the Langtang catchment in the Himalayas (Nepal). The paper is well-written and follows a clear line of argumentation. The approach of including as much as possible local and satellite data has a lot of merit. At the same time, I have major concerns about the methodology as detailed in the following.

We appreciate this positive feedback and below we explain our choice for the methodology and we will cover the concerns outlined by Anonymous Referee #3.

In my opinion, climate change scenario calculations based on simple (parameterized) snow models are unreliable as they necessarily present an extrapolation beyond the state for which the models have been calibrated. The problem with such simple snow models has been exemplarily shown by Magnusson et al. (2011). In this publication, a physics-based model and a model similar to seNorge are shown to produce similar results for a current climate but very diverging results for climate change scenarios.

We agree that in a perfect world a full energy balance model driven by observational data would be the ideal basis for a climate change impact study. However, the core of our paper is to show how a smart integration of remotely sensed snow cover imagery, data assimilation and a snow model can result in improved spatial estimate of snow water equivalent. We show the benefit of assimilating snow cover and snow depth into a snow model that simulates SWE and snowmelt runoff. This data assimilation approach would not have been computationally feasible in a physically-based model, due to the extensive parameterization and high number of dependent state variables. Previous approaches for snow(melt) modelling in the Himalaya dominantly relied on modelling a melt flux from a snow covered area (lacking information of SWE; e.g. Bookhagen and Burbank, 2010; Immerzeel et al., 2009) or used an inversed melt approach (Wulf et al., 2016) which only provides information on the maximum SWE for a snow season. This study is novel as both the snow water equivalent and snowmelt runoff of a Himalayan snowpack is explicitly simulated. This novel approach then also allows assessing changes in SWE and snowmelt runoff as result of changes in temperature and precipitation. This study is not intended to be a full-fledged study on climate change impacts as the data set is short and detailed information on changes in temperature and precipitation (patterns) is lacking (as already pointed out by RC1). In addition we are aware of the limitations regarding the use of parameterized snow models for climate change scenarios. Therefore we present climate sensitivity tests, showing the sensitivity of SWE and snowmelt runoff to changes in temperature and precipitation. This gives additional information about the sensitivity of the Himalayan snowpack under a changing climate, without stressing the use of a parameterized snow model for purposes that it is unsuitable for.

We agree that in particular the title may have given the reviewer the wrong impression about the focus of our study and we will modify the title to: 'Assimilation of snow cover and snow depth in a snow model to estimate snow water equivalent and snowmelt runoff in a Himalayan catchment'. In addition we will make it clearer in the manuscript that it is not a climate change impact study but merely a sensitivity experiment. We will also include a paragraph in the introduction on physics based versus "simple" snow models and we will include the Magnusson et al. (2011) reference.

The data assimilation via Kalman filtering potentially makes the modelling in the presented paper even more vulnerable to extrapolation than when using robust standard parameters. This is a major objection I have towards the methodology.

We agree that the Ensemble Kalman Filter (EnKF) has the potential to increase the vulnerability of a snow model to extrapolation as the model could be parameterized to fit the current climate and not future climate. However, in addition to our previous reply, we believe that this will be limited in this particular case because the posterior parameter distribution shows plausible values for the parameters. Using the EnKF actually provides valuable insight in the parameter and simulation uncertainty, compared to a deterministic simulation. In our case the narrow posterior distribution of parameter values together with plausible values shows the robustness of the parameterized snow model.

What is aggravating the problem described above is that the paper appears to completely ignore a large body of literature, which is based on physics-based snow modelling of climate change impacts. This has already been pointed out by RC1. I do not want to necessarily suggest that a paper based on temperature index modelling of future climate needs to be rejected in all cases. But if such an analysis is retained it needs to show a very careful assessment of potential errors through extrapolation and a discussion and comparison with results obtained with physics-based models.

We refer to our reply about the focus of our paper and we do not see where reviewer 1 indicated that physical-based snow models are a prerequisite for climate change impact studies, however we agree that a more thorough review regarding the pros and cons of physical-based models and temperature index models would improve the manuscript and provides a better context for our choices. In the revised manuscript a more extensive review will be given in the introduction about different model approaches (physical-based vs temperature index) and the results from previous studies. In the discussion the potential errors arising from non-stationarity and the extrapolation by a parameterized model will be addressed and placed in a proper context.

Computational restrictions do no longer prevent physics-based models to be applied to larger areas and for significant climate change studies. A recent example is Marty et al. (2017), which has just appeared in TC and which is a good starting point for the authors to find additional studies, which they need to discuss in context of their analysis.

Indeed there are no longer computational restrictions that prevent the use of physics-based models for larger areas. However, it is the limited data availability in the Himalayas that constrains the use of physics-based models. Physics-based models require extensive input data that is often unavailable in the Himalayas. Therefore it is necessary to use a parameterized snow model (requiring less input data) to perform SWE and snowmelt analysis of a Himalayan snowpack. To the authors' knowledge there is currently no study available that simulates the snowpack spatially distributed with a physics-based model for a Himalayan catchment. This supports our choice for a simpler approach.

We thank the referee for providing this interesting study. It will definitely help us to find more literature and it will enable us to put our results in a better context.

Interestingly, the results of the latter study (for the Alps) qualitatively agree with what the authors find for Langtang and this is a good sign. But this also means that the results are qualitatively not new and quantitatively highly uncertain for the argument presented above.

The same qualitative results show the capability and potential of the parameterized snow model to simulate climate sensitivity of SWE and snowmelt runoff. We disagree that these results are not new. To our opinion assimilating snow depth and remotely sensed snow cover into a snow model with parametrizations on melt modelling, albedo decay, avalanching, and snow compaction, and its first time application in a remote Himalayan catchment with the aim to understand the spatial patterns and climate sensitivity of SWE is quite novel.

This is my major point about the paper and I otherwise agree with the points raised by RC1.

**See the replies to the points raised by RC1.**

In general, presentation, figures and form of the paper are already at a very advanced state and almost without problems.

**References:**

Magnusson, J., Farinotti, D., Jonas, T. and Bavay, M. (2011), Quantitative evaluation of different hydrological modelling approaches in a partly glacierized Swiss watershed. Hydrol. Process., 25: 2071–2084. doi:10.1002/hyp.7958

Marty, C., Schlögl, S., Bavay, M., and Lehning, M.: How much can we save? Impact of different emission scenarios on future snow cover in the Alps, The Cryosphere, 11, 517-529, doi:10.5194/tc-11-517-2017, 2017.

**References:**

Bookhagen, B. and Burbank, D. W.: Toward a complete Himalayan hydrological budget: Spatiotemporal distribution of snowmelt and rainfall and their impact on river discharge, J. Geophys. Res. Earth Surf., 115(3), 1–25, doi:10.1029/2009JF001426, 2010.

Immerzeel, W. W., Droogers, P., de Jong, S. M. and Bierkens, M. F. P.: Large-scale monitoring of snow cover and runoff simulation in Himalayan river basins using remote sensing, Remote Sens. Environ., 113(1), 40–49, doi:10.1016/j.rse.2008.08.010, 2009.

Wulf, H., Bookhagen, B. and Scherler, D.: Differentiating between rain, snow, and glacier contributions to river discharge in the western Himalaya using remote-sensing data and distributed hydrological modeling, Adv. Water Resour., 88, 152–169, doi:10.1016/j.advwatres.2015.12.004, 2016.

---

## Editor Decision (ED1)

[revised manuscript text omitted]

---

## Author Response (AR2)

Dear Editor,

We are happy to submit our revised manuscript. The minor revisions proposed by the two referees greatly helped improving the current manuscript. Below we address the comments of the referees point-by-point (in *italics*) and we show our changes made to the manuscript in a marked-up manuscript version.

We are looking forward to your decision,

Best regards,

Emmy Stigter
(on behalf of all co-authors)

**Referee 1**

Review of the revised version of the paper (Referee 1) "Assimilation of snow cover and snow depth into a snow model to estimate snow water equivalent and snowmelt runoff in a Himalayan catchment" by E. Stigter et al. submitted to The Cryosphere.

The authors have proposed a clear and detailed answer to all my points and to the points raised by the two other referees. The quality of the manuscript has been increased during the reviewing process.

I have a remaining concern about the choice made by the authors to change the title of the paper. Indeed, the authors have decided to insist more on the core of the study : the assimilation of snow depth and snow cover in a snowpack model to improve the quantification of SWE and snowmelt runoff. The climate section is now presented as a sensitivity experiment and the authors explicitly stated in their answer that this paper is not a climate change impact study. This is supposed to justify the choice of the simple delta method used in the climate sensitivity tests.
If the core of the study is the assimilation of snow depth and snow cover in a snowpack model, the authors should insist more on this aspect in their paper. For example, it should be more emphasized in the abstract and in the conclusion. So far, the conclusion has only been slightly changed compared to the initial version of the paper and it seems to me that the main conclusions of the paper still concern the climate sensitivity experiment.

*We agree with anonymous referee 1 and both the abstract and conclusion are adapted so that the focus is more on the data assimilation.*

Since the main focus of the paper is the data assimilation of snowpack data, I also recommend the authors to add the following additional information:

- The impact of data assimilation on SWE and snow melt simulation could be also evaluated using the surface temperature data as an indirect evaluation. This would allow the authors to compare the presence of snow in the simulation and in the observation. The two transects for surface temperature cover a large altitudinal range and two different slope aspects (north and south). This would be an interesting complement to the spatial evaluation that has been carried out with the satellites data (MODIS and LandSat).

*These surface temperature observations have been used to validate the remotely sensed snow cover used in this study (MODIS and Landsat 8). This validation approach shows high classification accuracies for these snow cover maps (Section 3.1.2) and therefore shows that the snow cover maps are very suitable to evaluate/validate the simulated snow cover. We subsequently use the snow maps to validate the simulated snow cover spatially distributed. Altitudinal gradients and differences between northern and southern slopes are already captured in the remotely sensed snow cover. Hence using the surface temperature observations to validate the simulated snow cover is to our opinion an overlap in information.*

- The assimilation of snow depth data is carried out at two stations including Kyangjin station where almost no snow is found in 2014. Snow depth observations are used to optimize the parameter C6 governing snow settling in the model. Therefore, the impact of assimilation of snow depth from Kyangjin station is expected to be quite low in 2014. Time series of observed snow depth on Figure 5 suggest that Yala Pluvio or Yala BC could be used to carry out the assimilation of snow depth measurements. Since snow is present at these two stations in 2014, it could bring interesting improvements in model results. At least, the authors should comment the impact of the choice of stations for the assimilation of snow depth data.

*Only four sites with snow depth observations are available for both assimilation and independent validation. Two are selected for assimilation and two for validation. Station Kyangjin is specifically chosen for the assimilation as the snowpack is very dynamic (when present). There are indeed only very short moments of snow cover in 2014, whereas this is a longer time period in 2013. Once a snowpack forms it melts within only few days in 2014. The quick changes in the snowpack are useful for the assimilation. Therefore we think that station Kyangjin is a good station for assimilation of snow depth. Ideally more snow depth measurements would have been assimilated, but also stations were necessary for independent validation. We agree though that the choice of stations for snow depth assimilation might influence the results. This is now included in the discussion (p.14 l.20-21).*

- A discussion part on the benefits and challenges of assimilation of snow data (both spatial and punctual) would be very valuable. It could include a discussion around the choice of the EnKF framework compared to the particle filter that has been recently used in several studies on data assimilation for snowpack model (Charrois et al. 2016, Magnusson et al. 2017). The fact that the EnKF required continuous values (and not binary values) should be mentioned in the paper. It illustrates a limitation of the EnKF for the assimilation of spatially distributed snow data. Ideally, the assimilation of snow cover would be done on a pixel to pixel basis as mentioned by the authors in their answer.

*Although we agree with the reviewer that the implementation of a particle filter would be a good alternative, we purposely selected the EnKF for this work. As we deal with continues values, it is computationally efficient and allows for dual state-parameter estimations. The lower number of ensemble members compared to a particle filter allowed us to run multiple simulations over longer time periods, providing a better estimate of the potential of the EnKF improvements.*

*Both the particle filter and EnKF are similar in their approach (estimate the model uncertainty from the particle or ensemble spread), the EnKF has a higher efficiency when it deals with Gaussian data and related errors. The computational demand required for a particle filter exceeds the EnKF's computer requirements, due to the need to cover the entire (non-Gaussian) distribution. When the number of particles becomes too low, there is an additional risk of particle collapse, especially when one wants to take into account all the grid cells in the simulation with or without snow. This would require a total particle number exceeding the total number of grid cells in the domain, in combination*

*with all the possible parameter combinations to avoid collapse of the filter. For single site or small sites a particle filter would be a good alternative, but limited by the current available computational power, this is only feasible with an EnKF implementation.*

*We included the choice for the data assimilation scheme in the discussion along with the advantages and disadvantages (Section 3.2).*

[revised manuscript text omitted]